# The tumor suppressor TMEM127 regulates insulin sensitivity in a tissue-specific manner

Subramanya Srikantan[1], Yilun Deng[1], Zi-Ming Cheng[1], Anqi Luo[1], Yuejuan Qin[1], Qing Gao[1], Glaiza-Mae Sande-Docor[1], Sifan Tao[1], Xingyu Zhang[1], Nathan Harper[2], Chris E. Shannon[3], Marcel Fourcaudot[3], Zhi Li[4,10], Balakuntalam S. Kasinath[5], Stephen Harrison[6], Sunil Ahuja[2,7], Robert L. Reddick[8], Lily Q. Dong[4], Muhammad Abdul-Ghani[3], Luke Norton[3], Ricardo C.T. Aguiar[1,7,9] & Patricia L.M. Dahia [1,9]*

Understanding the molecular components of insulin signaling is relevant to effectively manage insulin resistance. We investigated the phenotype of the *TMEM127* tumor suppressor gene deficiency in vivo. Whole-body *Tmem127* knockout mice have decreased adiposity and maintain insulin sensitivity, low hepatic fat deposition and peripheral glucose clearance after a high-fat diet. Liver-specific and adipose-specific *Tmem127* deletion partially overlap global *Tmem127* loss: liver *Tmem127* promotes hepatic gluconeogenesis and inhibits peripheral glucose uptake, while adipose *Tmem127* downregulates adipogenesis and hepatic glucose production. mTORC2 is activated in *TMEM127*-deficient hepatocytes suggesting that it interacts with TMEM127 to control insulin sensitivity. Murine hepatic *Tmem127* expression is increased in insulin-resistant states and is reversed by diet or the insulin sensitizer pioglitazone. Importantly, human liver *TMEM127* expression correlates with steatohepatitis and insulin resistance. Our results suggest that besides tumor suppression activities, *TMEM127* is a nutrient-sensing component of glucose/lipid homeostasis and may be a target in insulin resistance.

[1] Division of Hematology and Medical Oncology, University of Texas Health Science Center at San Antonio (UTHSCSA), San Antonio, TX 78229, USA. [2] Division of Infectious Diseases, University of Texas Health Science Center at San Antonio (UTHSCSA), San Antonio, TX 78229, USA. [3] Division of Diabetes, Department of Medicine, University of Texas Health Science Center at San Antonio (UTHSCSA), San Antonio, TX 78229, USA. [4] Department of Cellular Systems and Anatomy, University of Texas Health Science Center at San Antonio (UTHSCSA), San Antonio, TX 78229, USA. [5] Division of Nephrology, Department of Medicine, University of Texas Health Science Center at San Antonio (UTHSCSA), San Antonio, TX 78229, USA. [6] Radcliffe Department of Medicine, University of Oxford, Oxford, UK. [7] South Texas Veterans Health Care System, Audie Murphy VA Hospital, San Antonio, TX, USA. [8] Department of Pathology, UTHSCSA, 7703 Floyd Curl Drive, San Antonio, TX 78229, USA. [9] Mays Cancer Center, UTHSCSA, 7703 Floyd Curl Drive, San Antonio, TX 78229, USA. [10] Present address: Department of Nephrology, The Third Xiangya Hospital, Central South University, 138 Tongzipo Road, Changsha 410013 Hunan, China. *email: Dahia@uthscsa.edu

Although the mechanisms of insulin action have been extensively studied, how tissues communicate to control the actions of insulin in glucose homeostasis is not completely understood[1]. Characterization of the molecular components of this inter-tissue interaction is relevant to understanding mechanisms of insulin resistance. This is a significant step toward addressing the growing epidemic of clinically relevant complications of insulin-resistant states in humans, including type 2 diabetes and nonalcoholic fatty liver disease.

The *TMEM127* gene encodes a highly conserved transmembrane protein that has no ascribed function. We discovered inactivating *TMEM127* mutations in the neuroendocrine tumors pheochromocytomas and, rarely, in renal carcinomas[2]. TMEM127 colocalizes with intracellular endomembranes, including early and late endosomes and lysosomes[3], and loss of TMEM127 affects early to late endosomal vesicle progression, suggesting that TMEM127 may function in endocytosis and endosomal trafficking[2,4]. We recently showed that TMEM127 forms complexes with lysosomal components of the mTORC1 pathway[5], a critical regulator of cell homeostasis[6].

To understand TMEM127's contribution to whole-organism physiology we generated whole-body *Tmem127* knockout mice (*Tm-CMV-KO*). These mice have reduced adiposity and are protected from age- and diet-dependent insulin resistance. Mice

with liver-specific *Tmem127* deletion (*Tm-LKO*) are also insulin sensitive and have reduced fat mass; unlike *Tm-CMV-KO*, *Tm-LKO* have increased hepatic lipogenic potential. In contrast, adipose-specific *Tmem127* knockout mice (*Tm-AKO*) are insulin-resistant. Mechanistically, hepatic Tmem127 promotes hepatic gluconeogenesis and inhibits peripheral glucose uptake, whereas adipose Tmem127 inhibits both adipogenesis and hepatic gluconeogenesis. Both human and mouse liver TMEM127 expression correlate with states of insulin resistance and advanced hepatic steatosis, suggesting that *TMEM127* may be a relevant target for treating insulin resistance and its hepatic complications.

## Results

**Whole-body Tmem127 null mice are small with low adiposity.** TMEM127 is ubiquitously expressed in both mice and humans (Supplementary Fig. 1A)[3]. We generated Tmem127 knockout (KO) mice by crossing mice carrying floxed *Tmem127* alleles (*Flx*) with mice expressing Cre recombinase under a generic (CMV) promoter (*Tm-CMV-KO*, Fig. 1a)[2]. Loss of *Tmem127* was confirmed by real-time PCR (Supplementary Fig. 1B). Mating of heterozygous Tmem127 mice produced offspring with a modest decrease in the expected number of KO mice (Supplementary Table 1, *p* = 0.03) not detectable during embryogenesis

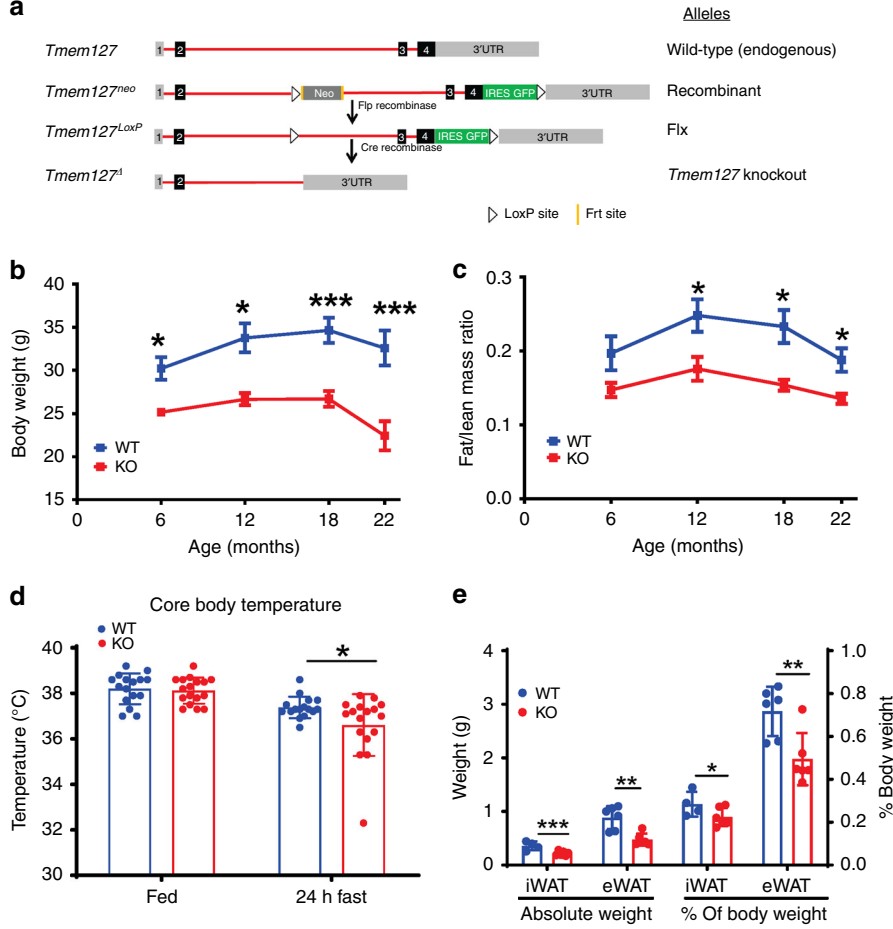

**Fig. 1** Global deletion of Tmem127 impairs growth and results in low adiposity. **a** Tmem127 recombinant mouse strategy; **b** body weight of male wild-type (WT) and Tmem127 knockout (KO) mice (*n* = 5–10 per genotype) from 6 to 22 months of age; **c** fat mass/lean mass ratio of male WT and KO mice across age (*n* = 5–10 per genotype); **d** core body temperature of adult WT and KO mice in the fed state and after 24 h fasting (*n* = 16–17 per genotype); **e** absolute weight of liver, inguinal white adipose tissue (iWAT), epididymal WAT (eWAT), or relative weight of these organs to total body weight in adult WT and KO mice (*n* = 4–6 per genotype); data were analyzed by Student's *t* test. Values are expressed as mean ± s.e.m. \**P* < 0.05; \*\**P* < 0.01; \*\*\**P* < 0.001. Adult mice were 9–12 months of age. See also Supplementary Fig. 1A–K and Supplementary Tables 1 and 2. Source data are provided as a Source Data file

(Supplementary Table 1), suggesting that loss of *Tmem127* reduces fitness at the perinatal/postnatal period. Surviving KO mice were viable and fertile. Growth rates of *Tm-CMV-KO* mice paralleled those of WT littermates, but KO males and females had lower body weight than WT littermates (Fig. 1b, Supplementary Fig. 1C) after birth (Supplementary Table 2). Both lean and fat mass were reduced (Supplementary Fig. 1D), but the fat mass was disproportionately affected in null mice, as seen by a significantly lower fat/lean ratio in both sexes (Fig. 1c, Supplementary Fig. 1E). The decrease in fat mass became more accentuated in adult animals, so we focused our experiments in the 9–12-month-old male mice.

**Thermogenesis is impaired in TM-CMV-KO mice.** In humans, germline loss-of-function *TMEM127* mutations confer susceptibility to neuroendocrine, sympathetic nervous system-derived tumors (pheochromocytomas) and renal cancers[2,3]. However, *Tm-CMV-KO* had no increased incidence of these, or other, tumors. Elevated catecholamines (epinephrine or norepinephrine), a hallmark of pheochromocytomas, increase lipolysis and contribute to weight loss in these patients[7]. However, we did not find high epinephrine or norepinephrine levels in plasma (Supplementary Fig. 1F) or adrenals of the KO mice. In addition, unlike our previous observation in pheochromocytoma patients[3], we found no evidence of mTORC1 target upregulation in the KO mice adrenals (Supplementary Fig. S1G). Thus, the decreased body weight and adiposity of *Tm-CMV-KO* were not the result of systemic hyperadrenergic levels or cancer-wasting effects.

Next, we evaluated caloric intake and energy expenditure, key contributors to reduced fat mass[8]. We found that *Tm-CMV-KO* food consumption was not significantly different from WT mice, but the KO had a small increase in resting metabolic rate (Table 1), consistent with higher energy expenditure. Two important components of energy expenditure are adaptive thermogenesis and physical activity[8]. To determine whether *Tm-CMV-KO* mice had enhanced thermogenesis, we examined their body temperature. Resting body temperature did not differ between WT and KO mice, but 24 h fasting resulted in a greater reduction in core body temperature in the KO mice (Fig. 1d). Moreover, serum levels of Fgf21, a hormone known to increase in fasting states and enhance lipolysis and thermogenesis[9], were significantly lower in the KO mice in the fed state and remained lower than WT after 24 h of fasting and 4 h after refeeding (Supplementary Fig. 1H). We also found no upregulation of the thermogenic marker uncoupling protein-1 in brown and white adipose tissues of KO mice (Supplementary Fig. 1I). Both the mass (Fig. 1e) and size (Supplementary Fig. 1K) of inguinal and epidydimal white fat (iWAT and eWAT, respectively) were markedly smaller in *Tmem127* null than in WT mice. Histologically, we observed reduced size of brown and white adipocytes, but no morphological aberrations (Supplementary Fig. 1J). We concluded that increased thermogenesis does not account for the higher metabolic rate of *Tm-CMV-KO* mice and,

in fact these mice have an impaired thermogenic response to fasting. Excessive physical activity can also account for reduced fat mass[10]. However, we found instead that *Tm-CMV-KO* mice displayed significantly reduced locomotor activity (Table 1).

**Tmem127 deletion impairs liver lipogenesis potential.** We next investigated fat synthesis, oxidation, and storage. iWAT (Fig. 2a) and eWAT (Supplementary Fig. 2A) showed no significant change in the expression of genes involved in fatty-acid storage, uptake, or oxidation pathways. Further, serum triglycerides (Supplementary Fig. 2B) and plasma-free fatty acids (Supplementary Fig. 2C) levels were not different from those of WT at regular feeding conditions. Calorimetry assessment did not show lower respiratory quotient in *Tm-CMV-KO* mice (Table 1). Taken together, these data suggest that the low adiposity was not due to augmented fat utilization as an energy substrate under regular fed conditions. In contrast, genes in the fatty-acid synthesis pathway (*Acaca*, *Acacb*, *Scd1*, *Elovl6*) were significantly upregulated in the iWAT (Fig. 2a) and eWAT (Supplementary Fig. 2A) of Tmem127 KO mice. In agreement with the mRNA data, expression of Fasn protein was higher in the *Tm-CMV-KO* (Fig. 2b), suggesting that the fatty-acid synthesis potential was not impaired, and might be instead augmented, in adipose tissue. Thus, our findings in fat tissue could not fully explain the low adiposity or increased metabolic rate of the *TM-CMV-KO* mice.

The liver plays a central part in fatty-acid synthesis and secretion for storage in adipose tissues during the fed state and, conversely, in lipid breakdown during starvation[11]. We found that both liver weight (Fig. 1e) and triglycerides levels (Fig. 2c) were decreased in *Tm-CMV-KO* mice. This was unlikely due to reduced fatty-acid supply to the liver[12], as we saw neither decreased free fatty acids (Supplementary Fig. 2B), nor increased Fgf21 levels in the serum of null animals (Supplementary Fig. 1H). Also, fatty acid oxidation genes were not upregulated in these samples (Fig. 2d). However, transcription of genes involved in lipid storage (*Cidea*, *Cidec*) and fatty-acid synthesis pathway were coordinately reduced in the liver of *Tm-CMV-KO* mice (Fig. 2d). Accordingly, protein levels of fatty-acid enzymes were decreased in these samples (Fig. 2e). We concluded that Tmem127 is involved in hepatic fatty-acid synthesis and storage, and that deregulation of these activities may contribute to the phenotype of the KO mice.

**Tmem127 deficiency leads to improved insulin sensitivity.** The low adiposity and reduced hepatic fat content suggested that adult *Tm-CMV-KO* mice might have an improved glucose metabolism. Indeed, we found significantly lower fasting blood glucose (Fig. 3a) and plasma insulin (Fig. 3b) levels in KO animals. Furthermore, the *Tm-CMV-KO* mice showed improved glucose tolerance (Fig. 3c) and maintained the ability to secrete insulin in response to glucose, albeit at lower levels than WT controls (Fig. 3d), despite comparable blood glucose levels (Supplementary Fig. 3A). Importantly, glucose disposal in the insulin tolerance

| **Table 1 Calorimetry features of adult *Tmem127* knockout mice** | | | | |
|---|---|---|---|---|
| **Measurement** | **WT** | **KO** | **Student's *t* test** | **ANCOVA** |
| RMR (mL/hr/glbm) | 1.636 ± 0.07 (*n* = 7) | 1.973 ± 0.128 (*n* = 6) | *P* = 0.05 | *P* = 0.09 |
| Food consumption (kcal/mouse/day) | 13.97 ± 1.5 (*n* = 16) | 12.16 ± 0.65 (*n* = 32) | *P* = 0.31 | *P* = 0.3 |
| Locomotor activity (beam breaks, total count) | 29,206 ± 2399 (*n* = 5) | 20,359 ± 1315 (*n* = 5) | *P* = 0.01 | *P* = 0.03 |
| RQ | 0.948 ± 0.014 (*n* = 7) | 0.961 ± 0.006 (*n* = 6) | *P* = 0.43 | *P* = 0.6 |

Data are presented as mean ± standard error of the means (SEM). Number of mice used in each experiment (*n*) is indicated. Measurements obtained from males, ages 9–17 mo under a regular diet
*RMR* resting metabolic rate, calculated from the five lowest values of the adjusted oxygen consumption ($VO_2$), measured in mL/hr/glbm (grams of lean body mass), *RQ* respiratory quotient, calculated by dividing carbon dioxide consumption ($VCO_2$) by $VO_2$ (*RQ*, $VCO_2/VO_2$), *ANCOVA* analysis of covariance

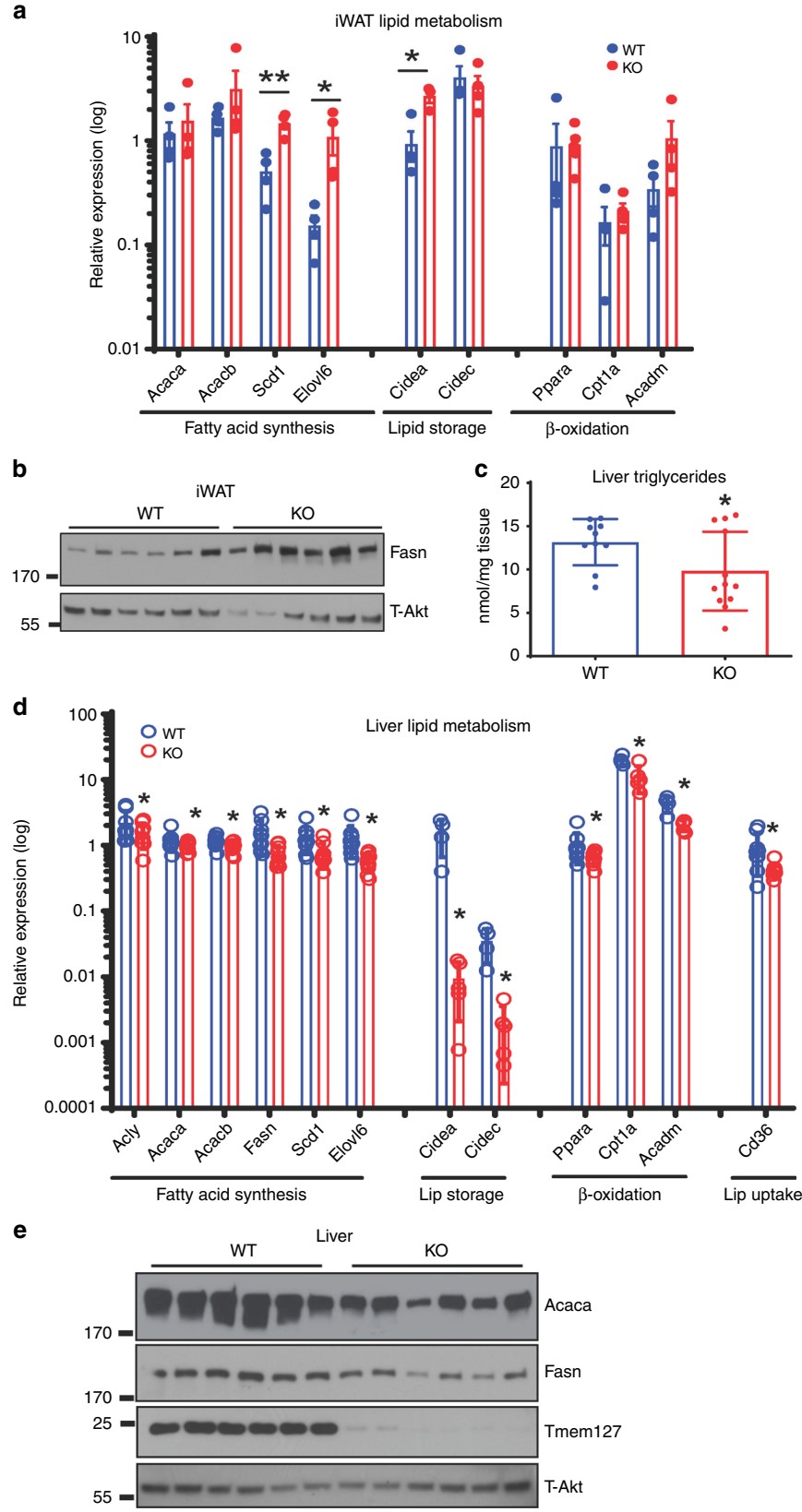

test was enhanced in the null animals (Fig. 3e). Together, the ability of *Tm-CMV-KO* mice to maintain normal glycemia in the presence of low insulin levels suggests that global *Tmem127* loss leads to increased insulin sensitivity in vivo, and the hypoinsulinemia might contribute to the lower body mass of these mice.

We found that both fasting glucose (Supplementary Fig. 3B) and insulin (Supplementary Fig. 3C) levels tended to be lower in the younger *Tm-CMV-KO* animals compared with age-matched controls. However, unlike adult mice, glucose (Supplementary Fig. 3D) and insulin (Fig. 3f, Supplementary Fig. 3E) tolerance

**Fig. 2** Global Tmem127 deficiency disrupts hepatic lipogenesis. **a** Relative iWAT mRNA expression (log) of the indicated fatty-acid synthesis, storage, oxidation, and transport gene expression in adult WT and KO mice (n = 4–5 per genotype) measured by real-time PCR (RT-PCR) and normalized to TfIIb gene; **b** western blots of fatty acid synthetase (FASN) and loading control (total AKT) in whole-tissue lysates prepared from iWAT from adult male WT or KO mice (n = 6 per genotype); **c** hepatic triglyceride content in fed adult WT and KO mice (n = 10–12 per genotype); **d** relative liver mRNA expression levels (log) of the indicated fatty-acid synthesis, storage, oxidation, and transport gene expression from adult WT and KO mice (n = 4–9 per genotype) measured by RT-PCR. **e** Western blots of Acetyl-CoA carboxylase (ACC), FASN and loading control (total AKT) in whole-tissue lysates prepared from liver from WT or KO adult male mice (n = 6 per genotype); data were analyzed by Student's t test. Values are expressed as mean ± s.e.m. *P < 0.05; **P < 0.01; ***P < 0.001. Adult mice were 9–12 months of age. See also Supplementary Fig. 2A–C. Source data are provided as a Source Data file

was indistinguishable from WT in the younger cohort despite the lower body weight of KO mice. Interestingly, *Tmem127* mRNA expression was higher in liver from older WT mice than younger animals (Fig. 3g). This profile suggests that Tmem127 expression might contribute to the age-related increase in insulin resistance.

The lower blood glucose of *Tm-CMV-KO* mice could be due to increased glucose uptake and metabolism by the liver and/or peripheral tissues, reduced hepatic gluconeogenesis or a combination of both. We measured the expression of genes in these pathways but found few changes in Glut4 expression or in the genes involved in glycolysis in WAT (Supplementary Fig. 4A), but a trend was noted toward increased *Pfkp* gene expression, suggestive of glycolysis, in muscle (Supplementary Fig. 4B) from *Tmem127* KOs. In the liver, *Tm-CMV-KO* had reduced expression of glucose transporter Glut2 and glycolytic (*Gck*) genes (Fig. 4a), consistent with reduced hepatic glucose flow. The response of *Tm-CMV-KO* to a pyruvate challenge was significantly lower than that of controls (Fig. 4b), suggesting reduced gluconeogenic potential, even though expression of genes involved in gluconeogenesis was not changed under fed conditions (Fig. 4a). To further evaluate gluconeogenesis, we subjected the *Tm-CMV-KO* mice to overnight fasting, a condition expected to trigger a hepatic gluconeogenic response[13]. *Tm-CMV-KO* mice displayed increased glucose clearance after overnight fasting (Fig. 4c). Moreover, the ability to induce liver transcription of *G6pase(G6pc)* and *Pepck1 (Pck1)* after 24-hour fasting was markedly attenuated in these mice (Fig. 4d). Based on these findings, we concluded that hepatic gluconeogenesis is diminished in mice lacking *Tmem127*.

We next analyzed liver lysates for phosphorylation of Akt, a key mediator of insulin signaling. Akt phosphorylation at serine 473 was higher in the liver lysates of KO mice (Fig. 4e). Serine 473 is phosphorylated by mTORC2[14], a critical mediator of insulin signaling and homeostasis[15,16]. Therefore, as a metric of mTORC2 activity[17], we examined the abundance of mTORC2 complexes (mTOR/Rictor) in the liver of *Tm-CMV-KO* mice and found them to be higher than in WT controls (Fig. 4f), suggesting that augmented mTORC2 might underlie the increased insulin sensitivity in the *Tm-CMV-KO* mice. In addition, insulin induced Akt phosphorylation was markedly higher in liver and muscle of null mice (Fig. 4g), in support of increased hepatic and peripheral insulin sensitivity.

**Tmem127 deletion protects from liver steatosis and insulin resistance.** We next asked whether Tmem127 deficiency would confer resistance to diet-induced obesity and its accompanying metabolic disruptions. We fed a cohort of WT and *Tm-CMV-KO* mice high-fat diet (HFD-60% kcal from fat) for 16 weeks. This regimen led to marked weight gain in both the WT and *Tm-CMV-KO* mice (Fig. 5a). In fact, the *Tm-CMV-KO* weight gain surpassed that of control animals, relative to their initial body weight (Supplementary Fig. 1A). This was predominantly due to fat, rather than lean, mass increase (Supplementary Fig. 5B), indicating that the ability to expand fat depots was maintained in the null mice. The body weight-adjusted food intake was not

significantly different between the two genotypes (Supplementary Fig. 5C), supporting higher feed efficiency (weight gain per calorie intake) of the null mice (Supplementary Fig. 5D). Although, similar to the WT controls, *Tm-CMV-KO* mice became glucose intolerant (Fig. 5b), their fasting insulin levels (Fig. 5c) and post-insulin glucose levels (Fig. 5d) were markedly lower than those of WT, suggesting that *Tmem127* loss protected mice from HFD-induced insulin resistance, independent of the weight gain.

To gather insights into the mechanisms underlying the insulin sensitivity of HFD-fed *Tm-CMV-KO* mice we examined their liver, fat, and muscle. All mice increased fat depot weight at the end of the HFD challenge (Fig. 5e), but the visceral fat (eWAT) expansion was greater in KO compared with WT mice (Fig. 5e). In contrast, both the absolute and relative liver weight were lower in *Tm-CMV-KO* (Fig. 5e). Concordantly, null mice had reduced hepatic fat deposition (Fig. 5f) and hepatic triglyceride levels (Fig. 5g). Moreover, the expression of lipogenic and lipid storage genes trended lower in the KO liver (Fig. 5h), similar to the pattern seen in chow-fed KO. Likewise, glucose uptake remained low, without significant upregulation of gluconeogenic transcription (Fig. 5i). These data suggest that despite the augmented insulin sensitivity, the capacity to synthesize fatty acids was impaired in the KO liver.

Conversely, and in keeping with the diet-induced fat mass increase (Fig. 5e), the expression of lipogenic and fat storage enzymes in fat tissue did not differ between null and WT animals (Supplementary Fig. 5E). However, a notable change was that both glucose transport and glycolytic enzyme expression were significantly increased in WAT from the HFD-fed null mice, indicative of increased glucose disposal (Fig. 5j). These data suggest that adipose expansion and increased glucose clearance in fat tissue contributes to the improved insulin sensitivity of the HFD-fed *Tm-CMV-KO*. Overall, these data are consistent with an effect of Tmem127 in promoting hepatic lipogenesis and inhibiting adipose glucose uptake and glycolysis.

**TMEM127 is required for hepatic lipogenesis transcription.** We hypothesized that reduced hepatic fat content contributes to the improved insulin tolerance of *Tm-CMV-KO* mice. Hepatic lipid synthesis is controlled by transcriptional activities and/or levels of lipogenesis-promoting transcription factors[18] and adipogenic factors[19,20]. We found that *Srebf1* mRNA expression in liver was similar between the control and null mice (Fig. 5K), but the expression of *Pparg*, *Cebpa* and, to a greater degree, *Chrebp*, were significantly downregulated in the *Tm-CMV-KO* liver, both under a regular diet and after HFD (Fig. 5k). HFD-fed KO mice also had significant decrease in both *Lxrα* (*Nr1h3*) and *Lxrβ* (*Nr1h2*), suggesting a coordinated downregulation of the transcriptional lipogenic program in the KO mouse liver (Fig. 5k). In contrast, the expression of these transcription factors was not significantly changed in iWAT or muscle (Supplementary Fig. 5G), suggesting a liver-specific defect. Taken together, our results point to reduced transcriptional input as a possible driver of diminished hepatic lipogenic capacity of the *Tm-CMV-KO* mice.

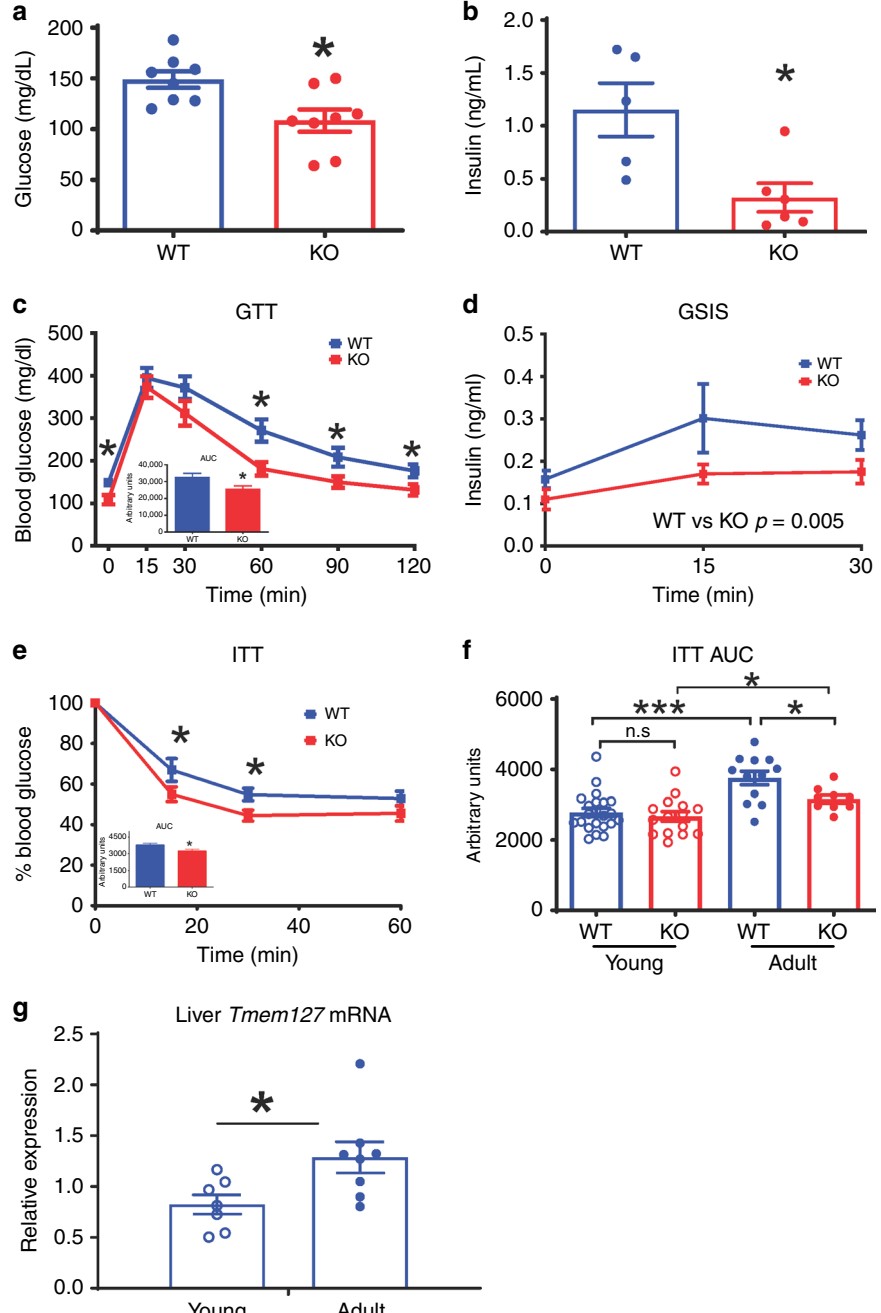

**Fig. 3** Loss of Tmem127 leads to hypoinsulinemia and improved insulin tolerance. **a** Fasting (6 h) blood glucose of adult WT and Tmem127 KO male mice ($n = 8$ per genotype); **b** fasting (6 h) plasma insulin of adult WT and KO male mice ($n = 5$–6 per genotype); **c** glucose tolerance test (GTT) and area under the curve (AUC, inset graph) of adult male WT and KO mice ($n = 8$ per genotype,) after 6-h fasting; **d** glucose-stimulated insulin secretion (GSIS) of adult male WT and KO mice ($n = 4$–7 per genotype); **e** insulin tolerance test (ITT) and AUC (inset) of adult male WT ($n = 10$) and KO mice ($n = 13$) after 6-h fasting; **f** ITT AUC values of young (3–6 months, WT $n = 15$; KO $n = 22$) and adult (9–12 months, WT $n = 10$; KO $n = 13$ per genotype) male WT and KO mice; **g** real-time PCR (RT-PCR) hepatic Tmem127 mRNA expression from WT young (3–6 months; $n = 7$) and adult (9–12 months; $n = 8$) male mice. Data were analyzed by Student's $t$ test or two-way ANOVA (in the case of GSIS). Values are expressed as mean ± s.e.m. $^*P < 0.05$; $^{**}P < 0.01$; $^{***}P < 0.001$. Adult mice were 9–12-month-old and young mice were 2–6 months old. See also Supplementary Fig. 3A–E. Source data are provided as a Source Data file

**Tm-LKO and Tm-AKO mice have differential insulin sensitivity.** Based on the findings above, we sought to evaluate the tissue-specific impact of Tmem127 deficiency. We generated mice lacking *Tmem127* exclusively in the liver (*Tm-LKO*) or in fat tissue (*Tm-AKO*), by crossing *Tmem127* Flx mice with mice expressing Cre under the albumin or adiponectin promoter, respectively (Fig. 6a). Both mice were viable and tissue-specific deletion of *Tmem127* was confirmed by protein expression (Fig. 6b).

Body weight of these two mouse strains was not different from Flx controls at young ages (Fig. 6c), but adult *Tm-LKO* had lower fat/lean mass ratio, whereas adult Tm-AKO had increased fat/lean mass ratio (Fig. 6d). In addition, the *Tm-LKO* mice had lower fasting glucose (Fig. 6e) and insulin (Fig. 6f) levels relative to controls. In contrast, although *Tm-AKO* mice showed no difference in glucose levels (Fig. 6e), their fasting circulating insulin was elevated (Fig. 6f). *Tm-LKO* were more insulin

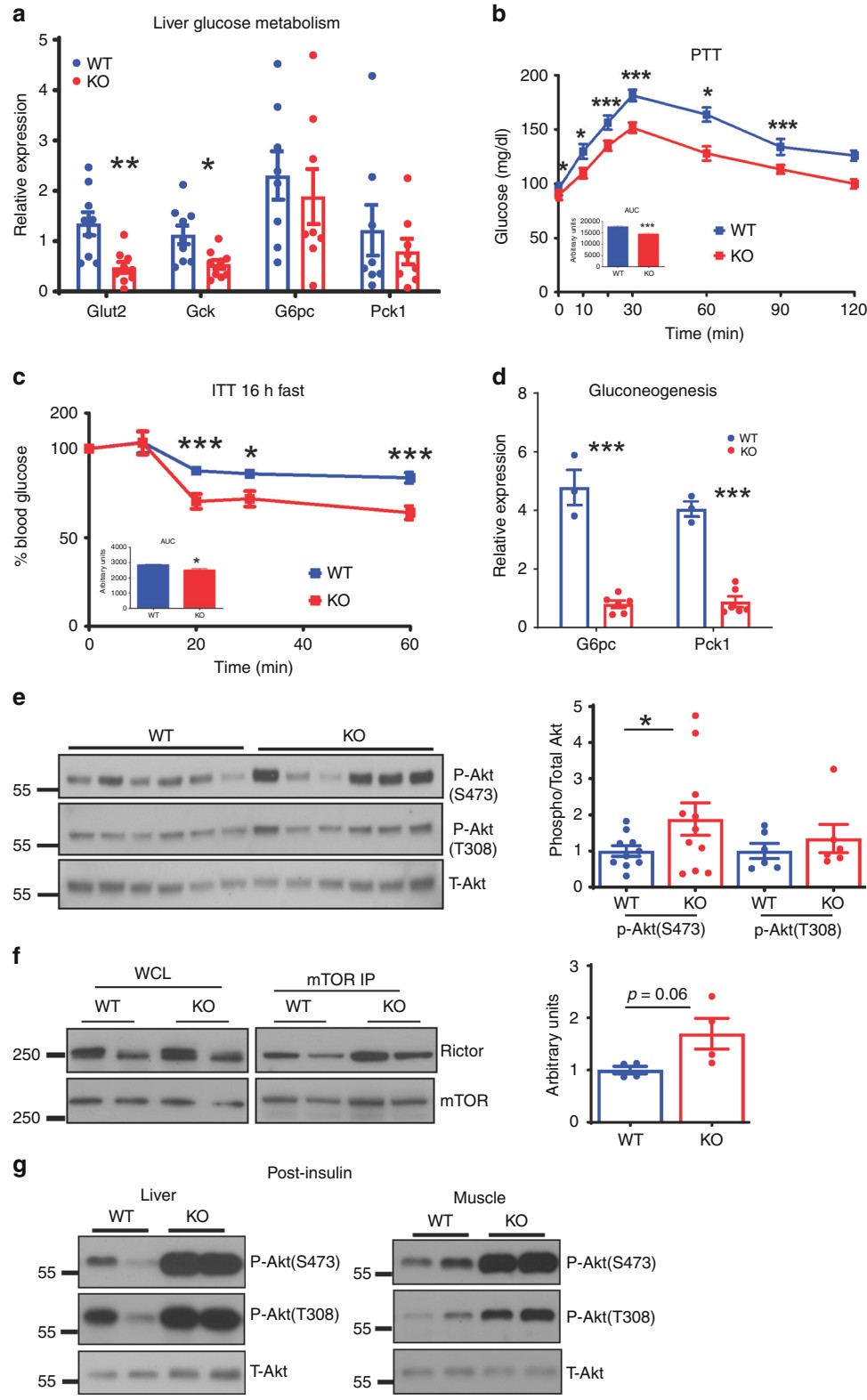

sensitive, even at a young age (Fig. 6g), whereas adult *Tm-AKO* developed insulin (Fig. 6g) and glucose (Fig. 6h) intolerance. Accordingly, the homeostatic model assessment of insulin resistance index (HOMA-IR) was higher in *Tm-AKO*, whereas that of *Tm-LKO* mice was lower, than controls (Supplementary Fig 6A). In keeping with this profile, *Tm-AKO* mice released more insulin in response to glucose, compared with controls

(Supplementary Fig. 6B). These observations suggest that *Tm-LKO* is insulin sensitive and *Tm-AKO* is insulin-resistant.

**Tm-LKO has increased insulin signaling**. Under chow-fed conditions, glucose transporter genes were upregulated in liver, muscle, and adipose tissue of the *Tm-LKO* (Fig. 7a), as was gly-colytic gene *Pfkp* (Fig. 7a). In addition, both liver and fat tissue

**Fig. 4** Tmem127 deletion decreases hepatic gluconeogenesis and increases insulin signaling. **a** Relative hepatic mRNA expression of the indicated glucose transporter, glycolysis and gluconeogenesis genes from adult WT ($n = 8$) and KO ($n = 9$) mice; **b** pyruvate tolerance test (PTT) and AUC (inset) of adult male WT ($n = 8$) and KO ($n = 9$) mice, after 16-h fasting; **c** ITT and AUC (inset) of adult male WT ($n = 6$) and KO ($n = 11$) mice after 16-h fasting; **d** RT-PCR hepatic *G6PC* and *Pck1* mRNA expression levels from 24 hour fasted (WT $n = 3$; KO $n = 6$) vs. fed (WT $n = 5$; KO $n = 9$) mice; **e** western blots of phosphorylated and total Akt proteins in liver whole-cell lysates from adult male, chow-fed WT, or KO mice, graph displays quantification of Akt phosphorylation at serine 473 or threonine 308 normalized by total Akt ($n = 6$ per genotype); **f** Rictor pulldown by mTOR immunoprecipitation (IP) in liver from adult male, chow-fed WT, or Tmem127 KO mice, corresponding whole-cell lysates (left panel), graph displays quantification of Rictor/mTOR IP complexes from two independent experiments and four mice per genotype (quantification performed by ImageJ and normalized to mTOR pulldown); **g** western blot of phosphorylated Ser473 or Thr308 Akt or total Akt from liver (left) or muscle (right) whole cell lysates of adult (>12 mo) male mice obtained after 6 h of fasting followed by injection of insulin (0.75 units/kg) or vehicle 15 min before harvesting ($n = 3$ mice per genotype per treatment condition, representative gel shown). Data were analyzed by Student's *t* test. Values are expressed as mean ± s.e.m. *$P < 0.05$; **$P < 0.01$; ***$P < 0.001$. Adult mice were 9–12 months of age. See also Supplementary Fig. 4A, B. Source data are provided as a Source Data file

had a trend toward higher AKT phosphorylation levels (Fig. 7b). These results support increased glucose flux and insulin signaling in these mice, phenocopying the *Tm-CMV-KO*. Since a feature of the latter was decreased fatty-acid gene/protein expression and concomitant coordinated downregulation lipogenic transcription factors, we next examined these pathways. However, unlike the *Tm-CMV-KO*, expression of several fatty acid synthesis genes (Fig. 7c) and lipogenic transcription factors (Fig. 7d) was instead upregulated in the liver of *Tm-LKO* and agreed with increased hepatic triglyceride levels in these mice (Supplementary Fig. 6C).

**Nutritional and insulin sensing are impaired in *Tm-AKO*.** During prolonged fasting, glucose is synthesized almost exclusively from gluconeogenesis[13,21]. We evaluated the gluconeogenic potential of *Tm-AKO* by measuring their hepatic gene expression after fed or prolonged fasting states. We found that hepatic gluconeogenic gene transcription in the fed state reached levels similar to those observed after fasting in the *Tm-AKO* (Fig. 7e), suggesting that baseline hepatic gluconeogenesis was not suppressed in conditions of nutrient supply, as in controls. In addition, expression of fatty-acid synthesis (Fig. 7f) and lipogenic transcription factors (Fig. 7g) were attenuated relative to control mice. These findings are consistent with higher hepatic gluconeogenesis and reduced hepatic lipogenesis under regular diet, supportive of hepatic insulin resistance, although we did not find appreciable differences in the levels of AKT phosphorylation (Supplementary Fig. 7A) or Rictor expression (Supplementary Fig. 7B) in *Tm-AKO* mice relative to controls under those conditions. In contrast, we observed high expression of Glut4 mRNA (Fig. 7h) and high levels of ACC protein (Fig. 7i) in adipose tissue, indicative of high glucose uptake in fat tissue and increased adipogenesis, respectively, both of which are associated with peripheral insulin sensitivity[22,23]. This agrees with the higher fat mass observed in the Tm-AKO (Fig. 6d). In indirect calorimetry, these mice displayed no change in resting metabolic rate (Supplementary Fig. 7C) but had reduced respiratory quotient (RQ, Fig. 7e), a measure of preferential use of fatty acids, as opposed to carbohydrates, as energy substrate. Collectively, these results suggest that adipose Tmem127 is relevant for the proper regulation of energy sensing, utilization and production. Although these data need to be evaluated under conditions of dietary stress and the contribution of other tissues still remain to be fully assessed, they support a model whereby *Tm-AKO* mice have adipose insulin sensitivity and hepatic insulin resistance.

**Cell autonomous and non-cell autonomous roles of TMEM127.** To determine cell autonomous and non-cell autonomous roles of *Tmem127* in the liver, we examined hepatic cells (human HepG2) lacking *TMEM127* via CRISPR-CAS9, engineered with previously reported guide RNAs[24] (Supplementary Fig. 8, Fig. 8a). HepG2

cells are derived from a well-differentiated human hepatoblastoma and retain characteristics of quiescent hepatocytes[25]. We observed that insulin induced AKT phosphorylation was increased in *TMEM127 KO* HepG2 cells (Fig. 8a). Furthermore, in response to a pyruvate challenge, the KO cells failed to induce gluconeogenic gene expression to the level of control cells (Fig. 8b), in favor of lower hepatic glucose production and similar to the in vivo profile of *Tm-CMV-KO*. In these cells, we also investigated the expression program of lipid synthesis and found that transcription of both fatty-acid synthesis genes and transcription factors followed the upward trend (Fig. 8c), though to a lesser extent, than that seen in the in vivo *Tm-LKO* model. To independently verify the cell autonomous effects of hepatic *Tmem127* loss, we examined primary hepatocytes derived from *Tm-CMV-KO* mice. These cells showed partial overlap with the fatty-acid synthesis gene expression seen in vivo, without reaching statistical significance (Fig. 8d). Taken together, our in vivo and in vitro models suggest that hepatic *Tmem127* has a cell autonomous role promoting hepatic gluconeogenesis but has less effect toward hepatic lipogenesis and therefore the reduced lipogenic potential of the whole-body KO liver is likely due to extra-hepatic (non-cell autonomous) effects.

**Tmem127 deficiency activates mTORC2.** As shown above, *Tmem127* loss results in increased mTORC2 activation (Fig. 4e–h and Fig. 8a). Several features of Tmem127 deficiency are anticorrelated with those observed in the liver-specific *Rictor* deletion model, the unique component of mTORC2 and a key mediator of metabolic homeostasis[13]. To further investigate the role of mTORC2 in the increased insulin signaling observed on the Tmem127 models, we knocked down *RICTOR* by shRNA in *TMEM127*-deleted HepG2 cells (Fig. 8e). In agreement with increased insulin signaling, cells null for *TMEM127* had higher AKT activation after insulin stimulation (Fig. 8e). However, *RICTOR* knockdown was able to downregulate mTORC2 targets in both control and TMEM127 KO cells. Interestingly, we observed higher levels of endogenous TMEM127 protein after *RICTOR* knockdown, suggesting that RICTOR and/or mTORC2 might regulate TMEM127 expression (Fig. 8e). Overall, our results suggest that RICTOR functions downstream of TMEM127 and support a reciprocal regulation of mTORC2 toward TMEM127 expression.

**Tmem127 expression correlates with insulin resistance states.** We noticed that TMEM127 expression increases after insulin stimulation in liver cells in vitro (Fig. 8a, e) and considered that this may be related to its role in vivo. Also, as shown above, hepatic Tmem127 was higher in adult mice when compared with young, more insulin-sensitive mice (Fig. 3g). Hence, we examined *Tmem127* expression in several mouse models of insulin

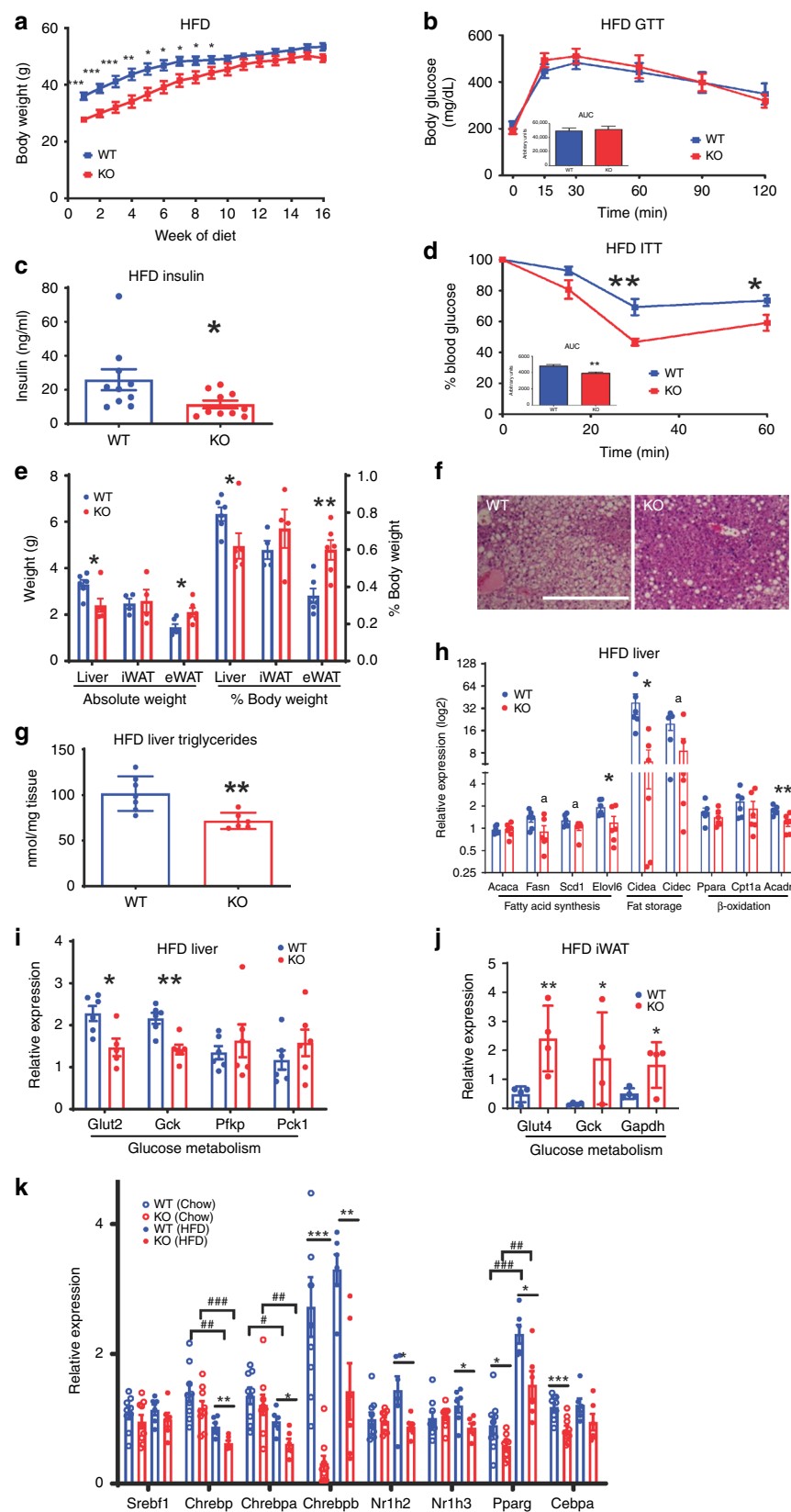

resistance. We found that liver *Tmem127* transcription increased after HFD in WT mice (Fig. 9a), but this change was reversed when these mice were placed on a regular diet for an additional 10 weeks (Fig. 9a). Interestingly, after HFD Tmem127 expression

in adipose tissue varied depending on the type of fat depot (Supplementary Fig. 9A, B), suggesting a complex regulation in adipose tissue. We next examined hepatic *Tmem127* mRNA in two well-known genetic models of diabetes, obesity, and insulin

**Fig. 5** Tmem127 loss protects against high-fat diet (HFD)-induced hepatic steatosis and insulin resistance. **a** Weekly body weight of WT and Tmem127 KO male mice on a 60% kcal fat diet for 16 weeks ($n = 7$–8 per genotype, starting age 5–7 months); **b** GTT and AUC (inset graph) of HFD-fed adult male WT ($n = 8$) and KO ($n = 10$) mice; **c** fasting (6 h) serum insulin of HFD-fed WT and KO male mice ($n = 10$ per genotype, 9–11 mo-old); **d** ITT and AUC (inset) of HFD-fed male WT and KO mice ($n = 6$ per genotype, 9–11 months); **e** absolute weight of liver, inguinal white adipose tissue (iWAT), epididymal WAT (eWAT), and respective relative weight of these organs to total body weight in HFD-fed WT ($n = 4$) and KO ($n = 6$) mice (9–11 months old); **f** representative hematoxylin and eosin (HE) sections of liver from HFD-fed WT and KO mice, scale bar is 500 μm; **g** hepatic triglyceride content in HFD-fed adult WT ($n = 6$) and KO ($n = 7$) mice (9–11 mo-old); **h** relative liver mRNA expression of the indicated fatty-acid synthesis, storage, oxidation, and transport gene expression from HFD-fed male WT and KO mice ($n = 6$ per genotype, 9–11 mo-old) measured by RT-PCR; **i** Relative liver mRNA expression of the indicated glucose transporter, glycolysis and gluconeogenesis genes from HFD-fed male WT ($n = 5$) and KO ($n = 6$) mice (9–11 mo-old); **j** relative iWAT mRNA expression of the indicated glucose transporter and glycolysis genes from HFD-fed male WT ($n = 4$) and KO ($n = 7$) mice (9–11 mo-old); **k** relative liver mRNA expression of the indicated lipogenic transcription factor genes from chow or HFD-fed male WT ($n = 6$ per diet) and KO ($n = 9$ per diet) mice (9–11 mo-old); Data were analyzed by Student's $t$ test or analysis of variance (ANOVA). Values are expressed as mean ± s.e.m. *$P < 0.05$; **$P < 0.01$; ***$P < 0.001$; a = $p = 0.06$. For 5 K: #comparisons between diet; * comparisons between genotype # or *$P < 0.05$; ## or **$P < 0.01$; ###or ***$P < 0.001$. See also Supplementary Fig. 5A–G. Source data are provided as a Source Data file

resistance, in which the metabolic disruption exists even under a regular diet, the D$b$/D$b$ and O$b$/O$b$ mice[26]. *Tmem127* expression was higher in the liver of homozygous D$b$/D$b$ (Fig. 9b) and O$b$/O$b$ (Fig. 9c) mice relative to their controls. We also evaluated liver of O$b$/O$b$ mice treated with pioglitazone, a thiazolidinedione with insulin sensitizing properties[27,28], and found that this treatment normalized *Tmem127* mRNA levels (Fig. 9c). Taken together, these findings support an association of *Tmem127* liver expression with states of insulin resistance. Importantly, these data also suggest that the increase in Tmem127 expression is reversible with the improvement of insulin sensitivity, either by diet (from HFD to chow) or by pharmacological intervention.

At last, to assess the translatability of our findings to human pathology, we examined liver biopsies from patients with histologically proven nonalcoholic fatty liver disease (NAFLD) or nonalcoholic steatohepatitis (NASH)[29]. In this cohort of 52 individuals, mean BMI was 31.3 kg/m[2] (range, 25.7–37.6) and twelve patients (23%) had diabetes. Twenty-four patients (46%) had NASH stage 1–3 fibrosis and 22 patients had NAFLD (stage 0–1 fibrosis). The diagnosis of NASH was defined as steatosis of > 20% of the liver and the presence of either hepatocyte ballooning or intralobular hepatocyte necrosis[29]. We found that hepatic *TMEM127* expression, measured by RNAseq, was higher in patients with liver disease relative to controls (Fig. 9d and Supplementary Table 3). This difference was highest when compared to patients with advanced disease, NASH ($p = 0.021$, Fig. 9d). Importantly, when other parameters were evaluated (Supplementary Tables 3 and 4), *TMEM127* counts correlated significantly with insulin levels ($R = 0.39$, $P = 0.005$, Fig. 9e) and HOMA-IR ($r = 0.35$, $P = 0.011$, Fig. 9f). Overall, these results are consistent with our observations in mice, and suggest that TMEM127 expression tracks with NAFLD/NASH in insulin-resistant patients.

## Discussion

In this study, we uncover the *Tmem127* tumor suppressor as an unsuspected regulator of insulin signaling and lipogenesis. Whole-body loss of *Tmem127* in mice results in low adiposity, improved glucose metabolic profile, and low insulin levels, which may contribute to the reduced body mass. Loss of *Tmem127* enhances insulin sensitivity and protects from diet-induced insulin resistance likely due to reduced hepatic gluconeogenesis and increased glucose clearance in fat tissue, although other mechanisms may also be involved. Moreover, these mice have low hepatic lipogenic gene expression, and we were able to show that this feature is not the primary determinant of their favorable insulin profile, as liver-specific *Tmem127* deletion also promotes insulin sensitivity but without reduced hepatic fatty-acid synthesis expression. This conclusion is corroborated by ex vivo data

showing that expression of lipogenic transcription factors or fatty-acid synthesis genes are not decreased in primary hepatocytes or in human hepatic cells null for *TMEM127*.

Our results point to a complex tissue-specific effect of Tmem127 on insulin sensitivity, with partially antagonistic effects in liver and fat (Fig. 9g). Although liver Tmem127 is likely involved in hepatic gluconeogenesis in a cell autonomous manner, adipose Tmem127 coordinates hepatic insulin regulation and adipogenesis. We also propose that the ability to respond to changes in nutritional status is impaired in mice lacking adipose tissue Tmem127 and this may account for the hepatic insulin resistance of this model.

We found that Tmem127 actions on insulin are independent of obesity. Indeed, the capacity of whole-body *Tmem127* null mice to remain insulin-sensitive despite diet-induced weight gain suggest that its deficiency may facilitate healthy adipose expansion after a dietary challenge, which in turn may lead to reduced hepatic fat deposition and improved hepatic and peripheral insulin sensitivity[30]. An examination of the effect of dietary stress in the liver-specific or fat-specific Tmem127 loss, and precise quantification of glucose uptake will be relevant to further shed light on insulin- and non-insulin-dependent tissue homeostasis. In addition, as neither liver nor adipose deficiency of Tmem127 fully recapitulate the phenotype of *Tm-CMV KO*, it is likely that other tissues involved in insulin homeostasis, including islet beta cells, muscle, hypothalamus, and macrophages, for example, might contribute to the metabolic outcome of global loss of *Tmem127*. Future studies should address their role in these metabolic pathways. Likewise, as our results support non-cell autonomous actions of Tmem127 loss, such as the effect of adipose Tmem127 on hepatic gluconeogenesis, it remains to be determined what factors, e.g., hormones, cytokines, metabolites, lipid composition, etc[31], mediate the tissue response.

The Tmem127 liver and fat loss profile is reminiscent of the complex actions of mTORC2 in these two tissues: loss of mTORC2-mediated by Rictor deletion in the liver results in age-dependent impairment of glucose metabolism owing to reduced Akt signaling[13,17]. Moreover, adipose mTORC2 regulates liver lipogenesis via modulation of Chrebp and promotes glucose uptake into adipocytes[22]. This mTORC2-mediated cross-talk between liver and adipose tissue leads to increased carbohydrate flux away from the liver under diet stress[22], strikingly similar to the *Tm-CMV-KO* profile after HFD (Fig. 5j). The mechanisms through which Tmem127 interfaces with mTORC2 remain to be elucidated. However, the observation that Rictor deficiency leads to accumulation of Tmem127 suggest possible feedback regulation, in support of an interaction between these two molecules.

We were able to exclude increased locomotor activity as a major contributor to the reduced fat mass and higher metabolic

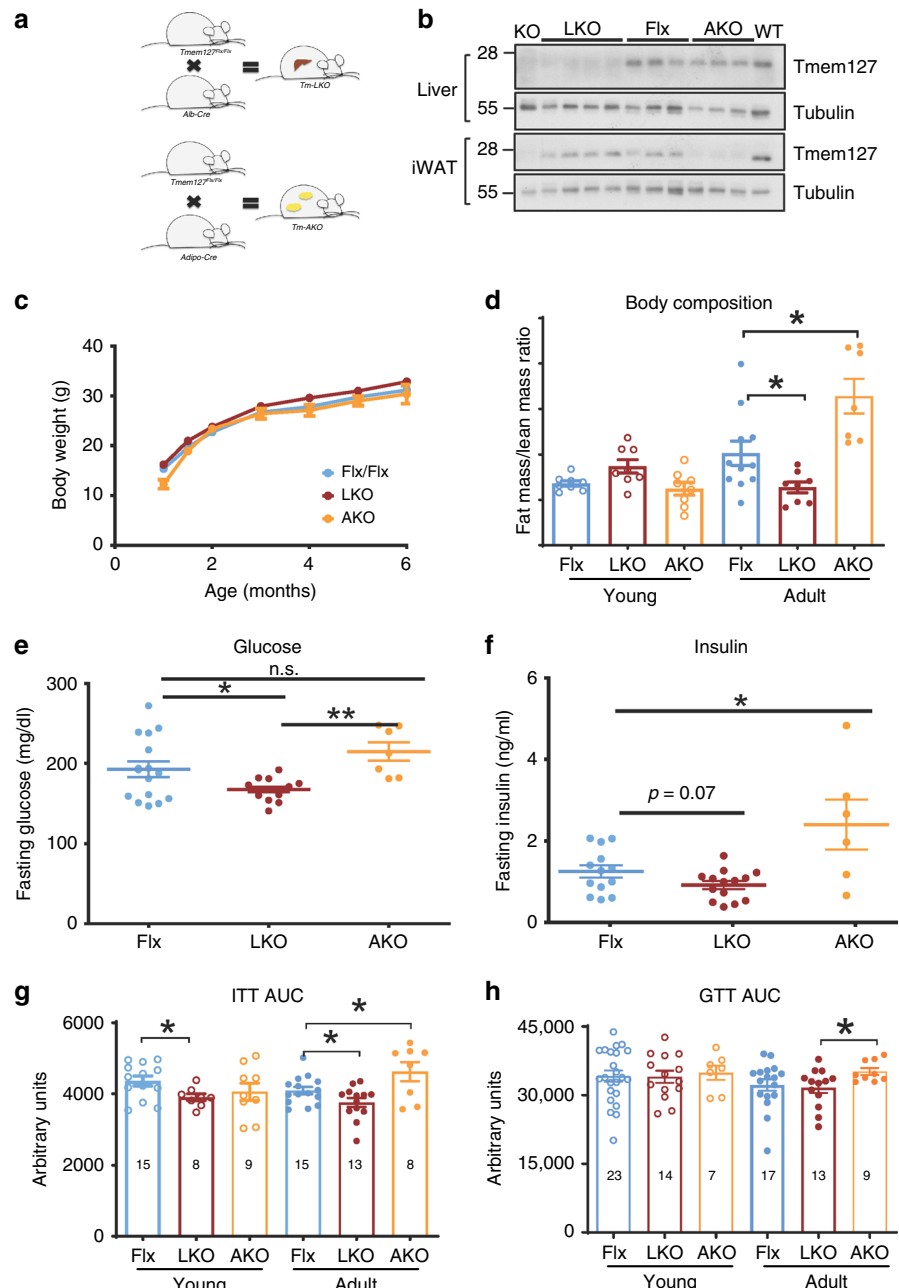

**Fig. 6** Tmem127 deletion in liver or adipose tissue leads to opposing glucose metabolism outcomes. **a** Strategy of generation of liver-specific (LKO) and adipose tissue-specific (AKO) Tmem127 deletion: Tmem127 Flx/Flx mice were crossed with Alb-Cre or Adiponectin-Cre mice, respectively; **b** western blot of liver or inguinal white adipose tissue (iWAT) lysates of Flx, LKO and AKO mice, alongside controls WT and KO probed with Tmem127 or a loading control antibody; **c** body weight of male Flx ($n = 16$), liver-specific Tmem127 KO (LKO, $n = 8$) and adipose-specific Tmem127 KO (AKO, $n = 8$) mice (from 2–6 months of age); **d** body composition of male Flx, LKO and AKO at 3–6 months of age (young; $n = 11$ LKO, 7 AKO, 7 flx) or 9–12 months old (adult; $n = 14$ Flx, 5 LKO; 7 AKO). Data are shown as lean/fat mass ratio. **e** Fasting blood glucose of adult WT and Tmem127 KO male mice ($n = 14$ LKO, 7 AKO, 16 flx); **f** fasting serum insulin of adult WT and KO male mice ($n = 13$ LKO, 7 AKO, 14 flx); **g** ITT AUC of young (3–6mo) and adult (9–12 mo) male Flx, LKO and AKO mice (number of mice per group is shown in each column); **h** GTT AUC of young (3–6 mo) and adult (9–12 mo) male Flx, LKO and AKO mice (number of mice per group is shown in each column). Data were analyzed by Student's t test; *$p < 0.05$; Values are expressed as mean ± s.e.m. *$P < 0.05$; **$P < 0.01$; ***$P < 0.001$. See also Supplementary Fig. 6A–C. Source data are provided as a Source Data file

rate of these mice. In fact, *Tm-CMV-KO* mice had decreased physical activity, similar to the phenotype of liver *Tsc1* KO mice, which have elevated mTORC1[32]. We previously reported that TMEM127 can antagonize mTORC1 signaling in human tumor samples and cell lines[2,3,5,33]. However, we did not find evidence of mTORC1 activation in the liver or hepatocytes lacking *Tmem127*. In the liver-specific *Tsc1* KO mice, mTOR-dependent induction of the insulin sensitizer Fgf21 mediates reduced locomotor

activity and augmented energy expenditure[32]. However, we found that *Tmem127* KOs had an impaired capacity to induce Fgf21. The mechanisms underlying this abnormal response and the increased energy expenditure of *Tmem127 KO* will need to be explored, but collectively they support an mTORC1-independent mechanism.

Other genes involved in oncogenesis, including *Pten, Akt, Bad*, also have key contributions to metabolism, and metabolic

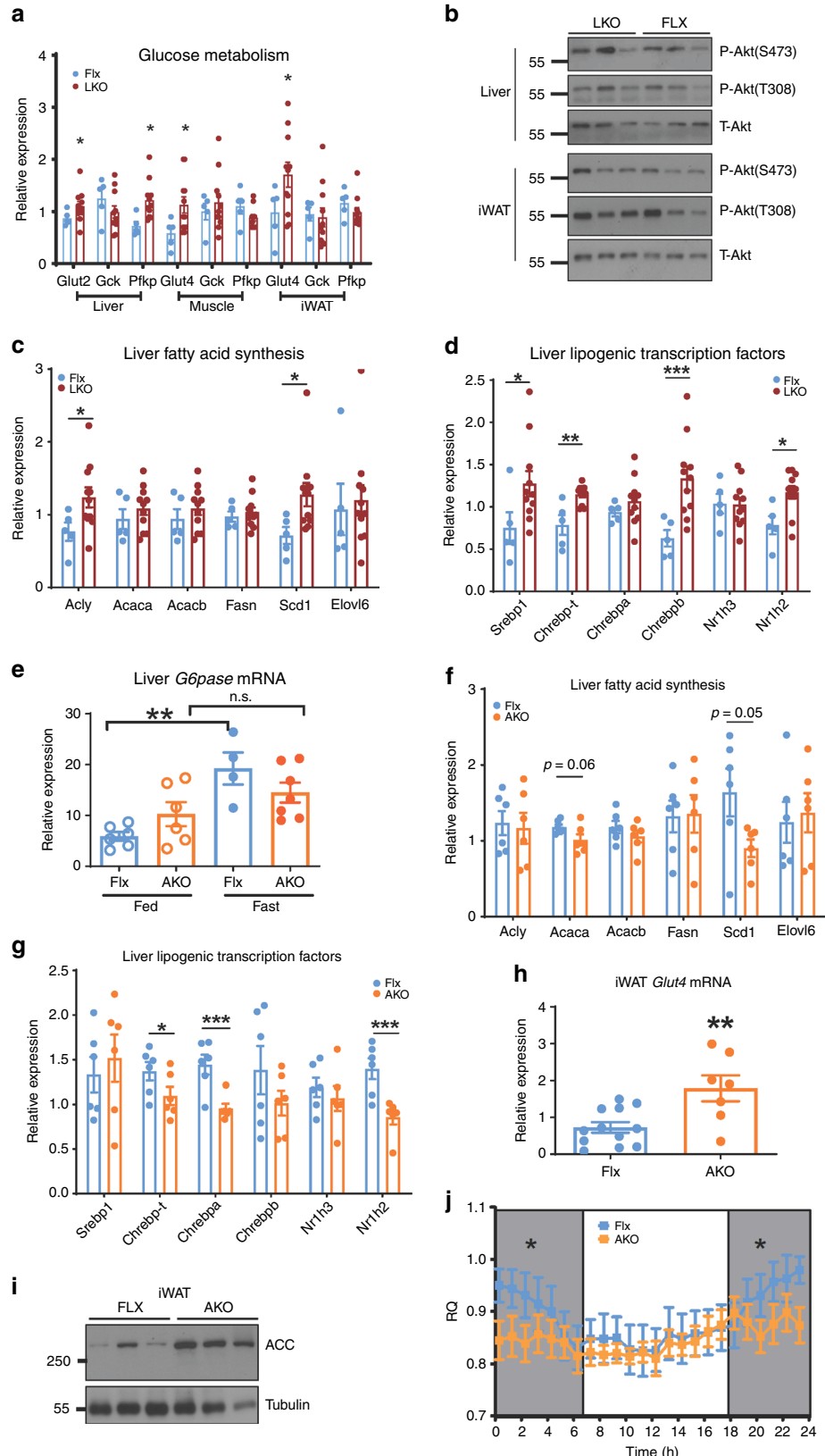

reprogramming is now recognized as a hallmark of cancer[34–36]. Future studies of *Tmem127* should integrate its tumor suppressor functions with its role in cellular glucose and insulin homeostasis. It is plausible that Tmem127 activities in endosomal trafficking and lysosome positioning may be linked to its ability to sense nutrients and modulate glucose metabolism[3,5]. Interestingly,

mTORC2 activity was recently shown to rely on lysosomal distribution[37], much like the mTORC1 dependency on a multi-protein lysosomal-anchored complex and its distribution in response to nutrients[38]. We previously identified TMEM127 associations with components of this mTORC1 lysosomal complex and effects on lysosomal positioning[5]. It remains to be

**Fig. 7** Liver-specific Tmem127 deletion increases hepatic and peripheral insulin sensitivity. **a** Relative liver, muscle, and iWAT mRNA expression of the indicated glucose transporter and glycolysis genes from Flx ($n = 5$) and LKO ($n = 11$) adult mice (9–12 mo old); **b** western blot of liver and inguinal white adipose tissue lysates from chow-fed LKO and Flx adult male mice, probed with phosphorylated Ser473 and Thr308 Akt and total Akt; **c** relative liver mRNA expression of the indicated fatty-acid synthesis genes from LKO and AKO adult male mice, represented as ratio over Flx ($n = 12$ Flx, 11 LKO and 7 AKO, 9–12 mo); **d** relative liver mRNA expression of the indicated lipogenic transcription factor genes from LKO and AKO adult male mice ($n = 11$ Flx, 12 LKO and 7 AKO, 9–12 mo). **e** Real-time (RT) PCR of liver G6pase gene expression in control or AKO 9–12 mo old male mice starved of food for 6 h and re-fed for 3 h (fed) or starved for 24 h (fasted), mice or feed condition (WT $n = 4$, KO $n = 7$); **f** RT-PCR of liver of the indicated fatty acid synthesis genes from chow-fed, adult male Flx ($n = 11$), and AKO ($n = 5$) mice; **g** RT-PCR of liver of the indicated transcription factor genes from adult male Flx ($n = 11$) and AKO ($n = 5$) mice; **h** RT-PCR of inguinal fat (iWAT) Glut4 gene expression from chow-fed, adult male Flx ($n = 11$) and AKO ($n = 5$) mice; **i** western blot of inguinal white adipose tissue lysates from chow-fed LKO and Flx adult male mice ($n = 3$ per genotype), probed with total ACC and a loading control; **j** respiratory quotient (RQ) of Flx ($n = 5$) or AKO ($n = 7$) chow-fed, adult male mice, data collected for 48 h and sorted by day or night cycle. Data were analyzed by Student's t test. Values are expressed as mean ± s.e.m. *$P < 0.05$; **$P < 0.01$. See also Supplementary Fig. 7A–C. Source data are provided as a Source Data file

determined if TMEM127 interacts with the mTORC2 complex at the lysosome and whether this putative interaction mediates any of the metabolic changes we observed in vivo.

Finally, and of translational relevance, we found that hepatic TMEM127 expression correlates with fatty liver disease and insulin-resistant states both in mice and in humans. Importantly, we saw that hepatic Tmem127 expression in mice is responsive to dietary or pharmacological interventions that ameliorate insulin resistance. These findings support TMEM127 as a potential target for insulin resistance which might be amenable to therapeutic modulation[35].

## Methods

**Generation of Tmem127 knockout mice models**. All animal studies were performed after receiving ethical approval of the University of Texas Health San Antonio (UTHSA) IACUC (Institutional Animal Care and Use Committee), and in compliance with all relevant ethical regulations for animal testing and research. Mice with recombinant Tmem127 allele (Tmem127$^{flx}$) were generated by crossing Tmem127$^{flx/+}$ mice[2] with CMV-Cre transgenic mice of C57BL/6 J background (Tg (CMV-cre)1Cgn; Jackson Laboratory)[39]. Tmem127$^{-/+}$ heterozygotes were mated to obtain WT and KO mice. For generating liver- and adipose-specific Tmem127 deletion, the Tmem127$^{flx/flx}$ mice were crossed with Albumin-Cre (Tg(Alb1-cre) 1Dlr; Jackson Laboratory)[40] and Adiponectin-Cre (Tg(Adipoq-cre)1Evdr; Jackson Laboratory)[41] transgenic mice, respectively. Tmem127$^{flx/flx;alb-cre/+}$ (referred to hereafter as TM-LKO) and Tmem127$^{flx/flx;adipoq-cre/+}$ (Tm-AKO) mice were compared with littermate 'Flx' controls (Tmem127$^{flx/flx;+/+}$). These strains are available under an MTA agreement. Leptin receptor knockout (Db/Db) and control (Db/+) as well as Leptin knockout (Ob/Ob) mice were purchased from Jackson Laboratories.

**Diet and treatment**. Mice were housed according to UTHSCSA guidelines, and procedures were approved by the Institutional Animal Care and Use Committee (IACUC), in compliance with the standards for the use of laboratory animals. Mice were fed standard rodent chow (Harlan) ad libitum unless mentioned otherwise. Mice on an HFD regimen were fed a diet containing 60% kcalories fat (D12492, Research Diets) for 16 weeks. A subset of this group was switched back to chow diet for 10 additional weeks. For the pioglitazone studies, 12-week old Ob/Ob mice were given ad libitum access to food containing pioglitazone for 4 weeks (diet was custom-made by Research Diets to contain 300 mg pioglitazone/kg diet, 10% kcalories fat). On average mice consumed 25 mg/kg/day of pioglitazone, which previously has been shown to significantly improve hepatic and adipose tissue insulin sensitivity[28].

**Body temperature, body composition, and energy expenditure**. Core body temperature was measured using a rectal probe connected to Animal Temperature Controller (#ATC2000; World Precision Instruments). Fat mass, lean mass, and percentage of fat were determined using Dual-Energy X-Ray Absorptiometry (GE Medical Systems). For energy expenditure measurement, mice were placed in metabolic cages, and energy expenditure was measured by indirect calorimetry using the MARS system from Sables Systems (Las Vegas, NV) for the Tm-CMV-KO, and the Columbus CLAMS system (Columbus, OH) for the Tm-AKO mice. Oxygen consumption ($VO_2$), carbon dioxide production ($VCO_2$), and RQ are measured for each mouse. Animals are habituated during the first 16 h and measurements and data recorded during the subsequent 24 h. Total activity and sleep behavior were measured by Open-Field activity monitoring. The number of beam breaks across the X–Y axis is measured to determine total activity. Sleep and sleep fragmentation assessments are based on a model developed by Pack et al for high-

throughput phenotyping of sleep in mice[42]. The technique uses activity/inactivity assessments as a measure of sleep, with sleep defined as any bout of inactivity of ≥ 40 s. Based on their results, we measure activity across a 24-h period, distinguishing between light and dark phases to assess sleep and sleep fragmentation.

**Human samples**. Samples and clinical data were obtained after signed informed consent in accord with ethical standards through Institutional Review Board-approved research repositories at the Brooke Army Medical Center and the UTHSA. A total of 52 individuals (20 males, 53.4%), ages 20–64 and average BMI of 31.3 (range, 25.7–37.6) were included in the study. Based on their liver histology, seven individuals with no liver disease were used as controls, 22 patients were classified as having steatosis with stage 0 fibrosis (NAFLD), and 24 patients had NASH stage 1–3 fibrosis, according to well-established histopathological criteria[29]. Twelve individuals (23%) had diabetes. Information available from these patients included gender, BMI, insulin, and glucose; HOMA-IR and NAFLD score were derived from these parameters (Supplementary Table 4).

**RNASeq**. Whole-genome transcriptomic analyses were performed on liver tissue obtained by biopsies from seven healthy controls, 21 patients with nonalcoholic fatty liver disease (NAFLD/NNN) and 24 patients with NASH (nonalcoholic steatohepatitis), of which 16 were stages 0/1, five were stage 2, and three were stage 3. Total RNA was isolated using the RNeasy Mini Kit (Qiagen, Hilden, Germany) with DNase I digestion according to the manufacturer's instructions. RNA quantity and purity were determined by spectrophotometry (260/280 = 1.8–2.0) (Nano-drop) and integrity (RIN) was determined using an Agilent 2100 Bioanalyzer with an RNA 6000 Nano assay (Agilent Technologies, Palo Alto, CA). Samples with RIN ≥ 7.0 were selected for RNA sequencing. Double-stranded cDNA library was prepared starting from 1 μg of total RNA input according to the TrueSeq RNA v2 sample preparation kit protocol (Illumina, San Diego, CA). In brief, mRNA was selected using poly-T oligo-attached magnetic beads and then fragmented. First and second cDNA strands were synthesized and end-repaired. Multiplexed adaptors were ligated after 3′-adenylation. Double-stranded cDNA templates were enriched by PCR. Libraries were validated using a DNA High Sensitivity assay on the Agilent 2100 Bioanalyzer (Agilent Technologies, Palo Alto, CA) and quantified by a Kapa Library quantification kit (Kapa Biosystems, Woburn, MA). Libraries were clustered using the Illumina cBot (Illumina, San Diego, CA) and then paired-end sequenced (2 × 101 bp) on an Illumina HiSeqTM 2000 (Illumina, San Diego, CA). Base calling and quality filtering were performed using the CASAVA v1.8.2 (Illumina, San Diego, CA) pipeline. Sequences were aligned and mapped to the UCSC hg19 build of the Homo sapiens genome (from Illumina igenomes) using tophat v2.0.1 7. Gene counts for 23,239 unique, well-curated genes were obtained using HTSeq framework v0.5.3P3 (http://www-huber.embl.de/users/anders/ HTSeq/doc/history.html). Gene counts were normalized, and dispersion values were estimated using the R package, DESeq v1.10.18.XX. A generalized linear model based on the negative binomial distribution with the likelihood ratio test was to test for associations between TMEM127 expression and: disease states (controls, NAFLD, and NASH), insulin levels, or HOMA-IR scores[43]. P value < 0.05 was considered significant. Both normalized and raw counts are provided as Supplementary Table 4 and Source files.

**Generation of TMEM127 knockout HepG2 cells with CRISPR/Cas9**. TMEM127-null HepG2 cells were generated by CRISPR-Cas9 technology. Guide RNAs targeting the first and last coding exon of the TMEM127 gene (ATGTAC GCCCCCGGAGGCGC and CTTCGCCGTTAGCTTCTACC, respectively) were cloned into pLentiCRISPRv2, BsmBI digestion, one vector system[33]. Cells were transduced with LentiCRISPRv2 expressing each guide or empty LentiCRISPRv2 as control, and single clones were selected with puromycin[44]. The KO status of these clones was verified by Sanger sequencing (Supplementary Fig. 8) and Western blot (Fig. 8a, e). Clone TMEM127 T2-2 was used for further experiments.

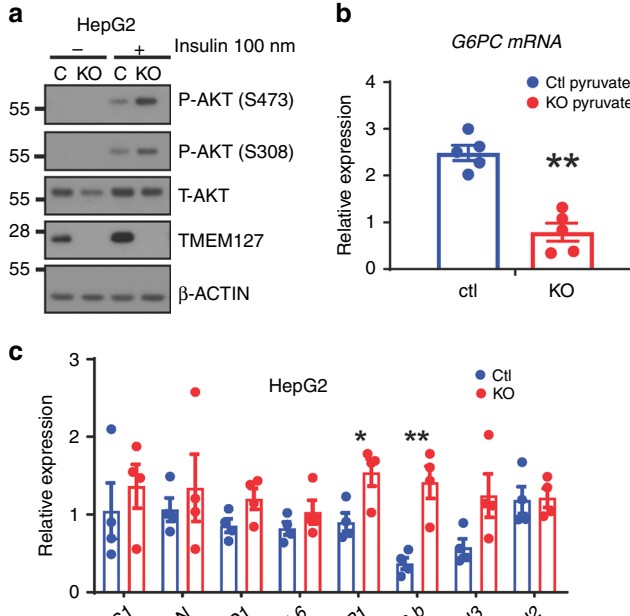

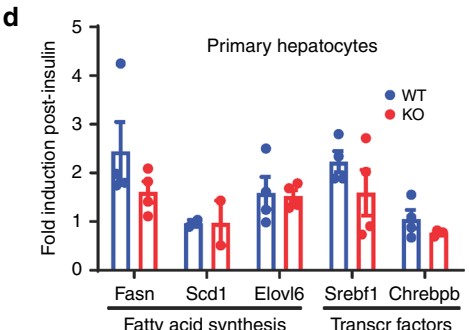

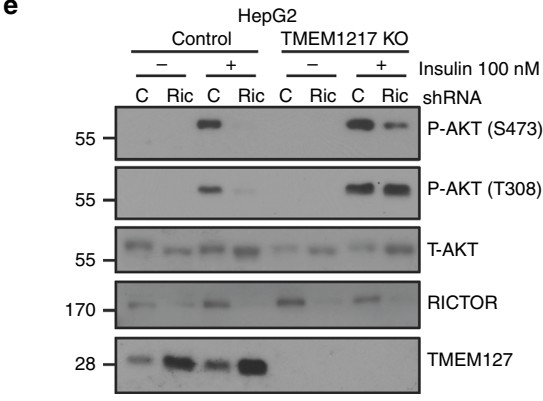

**Fig. 8** Effects of TMEM127 deletion in hepatic gluconeogenesis and lipogenesis are cell autonomous. **a** Western blot of lysates from control (C) or CRISPR-Cas9 mediated- TMEM127 knockout KO HepG2 human hepatic cells treated with or without 100 nм insulin for 10 min, probed with Akt, TMEM127 or a loading control antibody; **b** real-time (RT) PCR of *G6PC* (the G6Pase gene) expression in control or TMEM127 KO HepG2 cells deprived of glucose and stimulated with pyruvate 2 mм (three biological repeats); **c** RT-PCR mRNA expression of the indicated fatty-acid synthesis, transcription factors, and glucose transporter genes from HepG2 control or TMEM127 KO cells (three biological repeats); **d** RT-PCR of fatty-acid synthesis or transcription factors from primary hepatocytes from Tmem127 WT or KO (Tm-CMV-KO); 12-week old-female mice, *n* = 3 per genotype). **e** Western blots of indicated phosphorylated and total proteins from whole-cell lysates of HepG2 TMEM127 KO or control cells with or without Rictor knockdown (Rict) and treated with insulin as indicated. Shown is one of four replicate experiments. Data were analyzed by Student's *t* test. Values are expressed as mean ± s.e.m. *P < 0.05; **P < 0.01. See also Supplementary Fig. 8. Source data are provided as a Source Data file

**Cell culture, transfections, and transductions.** HepG2 cell line was obtained from ATCC (HB-8065), and mouse embryonic fibroblasts were isolated from Tmem127 mice using standard methods[2]. HepG2 cells were cultured in Eagle's Minimum Essential medium (MEM; #10-009-CV, Corning) supplemented with 10% fetal bovine serum and 100 U/ml penicillin and streptomycin (#30-002-CI, Corning). Primary hepatocytes were cultured in Williams medium supplemented with 10% fetal bovine serum and 100 U/ml penicillin and streptomycin[46]. All cells were negative for Mycoplasma infection when tested using PCR-based protocol[47]. HepG2 cells were transduced with lentivirus PTM-LKO.1 empty vector or pTM-LKO.1-Rictor-shRNA[33]. Transduction efficiency was verified by Rictor western blot. For insulin treatment, 80% confluent cells were serum starved for 16 h and challenged with 100 nм Insulin (#I9278; Sigma) for indicated time points and harvested for protein or RNA analysis.

**RNA isolation and qPCR analysis.** RNeasy kits (Qiagen) were used for RNA isolation from liver and cell lines (RNeasy Mini Kit), muscle (RNeasy Fibrous Tissue Mini Kit), and fat tissues (RNeasy Lipid Tissue Mini Kit) using the manufacturer's suggested protocol. One μg of total RNA was used for cDNA synthesis (High capacity cDNA synthesis Kit; Applied Biosystems). Quantitative PCR reactions containing the SYBR green mix (Applied Biosystems) were performed with StepOnePlus Real-time PCR System (Applied Biosystems). Triplicate runs of each sample were normalized to Tfiib (mouse tissues) or TBP (human, cell lines) to determine relative expression levels. Primer sequences are available in the Supplementary Information.

**Immunoblotting.** Liver and muscle were lysed in cold RIPA buffer (50 mм Tris-HCl pH 8.0, 150 mм NaCl, 1% NP-40, 0.5% Sodium deoxycholate, 0.1% Sodium dodecyl sulfate, 10% Glycerol) supplemented with phosphatase and protease inhibitor cocktails with mechanical agitation using a hand-held homogenizer. Lysates were incubated on ice for 30 min and sonicated for 30 s on ice. Fat tissue samples were lysed in cold HNTG buffer (50 mм HEPES pH 7.5, 150 mм NaCl, 10% glycerol, 1% TritonX100) with mechanical agitation and lysates incubated on ice for 30 min. Cells were lysed in cold RIPA buffer supplemented with phosphatase and protease inhibitor cocktails and mechanical agitation, followed by incubation on ice for 15 min. All lysates were centrifuged at 16,000 × g for 15 min, supernatant was collected, and protein concentration measured by Bradford assay (#500-0006; Bio-Rad). Twenty μg of protein was separated on 8%, 10%, 12% or 4–12% gradient Tris-Glycine gel by sodium dodecyl sulphate-polyacrylamide gel electrophoresis (SDS-PAGE). Alternatively, some cell lysates were prepared by direct lysis with 1× Laemmli buffer and separated on SDS-PAGE. Blots were probed with antibodies at the indicated dilutions: Acaca (3676, 1:10,000), Fasn (3180, 1:5000), Akt (9272, 1:1000), phospho-Akt,S473 (9271; 1:1000), phospho-Akt,T308 (13038; 1:1000), Rictor (9476; 1:1000), mTOR (2983; 1:1000), phospho-S6K,T389 (9206; 1:1000), S6K (9202; 1:1000), phospho-4EBP1,T37/46 (9459; 1:1000), from Cell Signaling Technologies; β-actin (A2228; 1:20,000) from Sigma-Aldrich, β-tubulin (ab6046; 1:10,000) from Abcam, and Tmem127 (A303-450A, 1:1000) from Bethyl laboratories. X-rays were scanned and quantified using ImageJ software (NIH). Uncropped and unprocessed scans of the most important blots are available in the Source Data file.

**Tissue lysis and immunoprecipitation.** Liver tissue was lysed in ice-cold CHAPs lysis buffer (0.3% CHAPS, 10 mм beta-glycerol phosphate, 10 mм sodium pyrophosphate, 40 mм HEPES (pH 7.4) and 2.5 mм MgCl2) with EDTA-free protease inhibitor (Roche #11836170001). Lysates were cleared by centrifugation at 16,000 g

**Primary hepatocytes.** Hepatocytes were obtained from 3–5-month-old-female mice (*n* = 4 for each genotype) by collagenase perfusion until complete digestion. Mice were euthanized during perfusion. Livers were transferred into Petri dishes containing ice-cold HBSS/HEPES buffer (370 mм NaCl, 5 mм KCl, 0.33 mм Na2HPO4, 0.44 mм KH2PO4, and 10 mм HEPES (pH 7.4)), minced, strained and washed with HBSS/HEPES by low-speed centrifugation (500 rpm at 4 °C for 2 min). Freshly isolated hepatocytes were suspended in Williams' medium, and viability (85–95%) was determined by trypan blue dye exclusion. Cells were plated on collagen type I-coated six-well plates at 2 × 10⁶/well[45]. After 4 h, medium was replaced with serum-free media overnight, and stimulated for 6 h with 10 nм insulin. Cells were harvested for RNA, reverse transcribed and used for quantitative real-time PCR as described below.

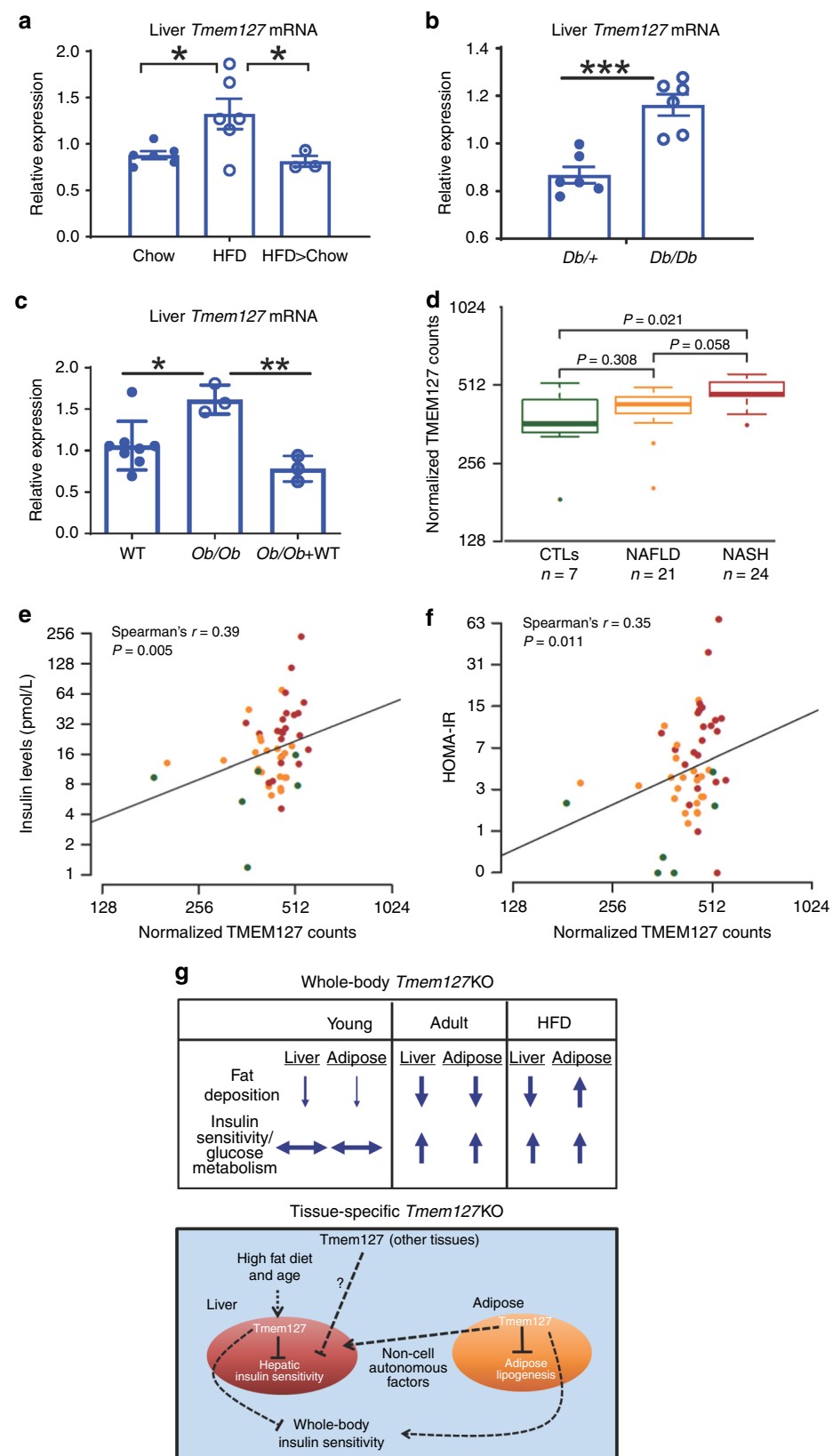

at 4 °C for 10 min. Cleared lysates were normalized for protein content by Bradford assay, and 2–3 mg of protein lysate were incubated with mTOR antibody at manufacturer's recommended dilutions at 4 °C overnight on a rotator, followed by addition of protein A agarose beads (Pierce #22811; 40 μl of 50% slurry per 1 mg of lysate) and continued incubation for 2 h at 4 °C. Beads were washed three times with CHAPs lysis buffer supplemented with 150 mM NaCl, eluted with 2 × Laemmli buffer by boiling at 100 °C for 10 min and separated on SDS-PAGE. Probed blots were quantified using ImageJ. Uncropped and unprocessed scans of the most important blots are available in the Source Data file.

**H&E staining and fat cell size measurement**. For H&E staining, adipose and liver tissues were fixed with a buffer containing 10% formalin for 24–48 hr and embedded in paraffin. Tissue sections (6 μm thick) were stained with H&E. Fat cell

**Fig. 9** Hepatic TMEM127 expression correlates with states of insulin resistance in humans and mice. **a** Relative liver Tmem127 mRNA expression from adult WT male mice under chow ($n = 6$), high-fat diet (HFD, $n = 6$) for 16 weeks or after HFD followed by 10 weeks of chow (HFD > chow, $n = 3$ per genotype) measured by RT-PCR; **b** relative liver Tmem127 mRNA expression from adult male *Db/db* or age-matched heterozygous db mice ($n = 6$ per genotype); **c** relative liver Tmem127 mRNA expression of control ($n = 8$), *ob/ob* ($n = 3$) or *ob/ob* ($n = 3$) adult mice treated for 4 weeks with the insulin sensitizer pioglitazone. **d** Log2-transformed, normalized *TMEM127* gene RNAseq read counts in liver biopsies of patients with the indicated status: CTLs = control individuals; NAFLD = nonalcoholic fatty liver disease; NASH = nonalcoholic steatohepatitis; the number of samples per group is indicated below the graph. The statistical test was the negative binomial generalized linear model implemented in DESeq with a likelihood ratio test to compare *TMEM127* expression between the groups. The thick horizontal line represents median values, box represents 25th and 75th percentiles, whiskers represent minimum and maximum values, outlier samples are shown as dots; **e** correlation between TMEM127 mRNA levels shown in **d** and insulin levels ($n = 52$). Symbols represent individual patients control samples are shown in green, NAFLD are orange and NASH are red; **f** correlation between TMEM127 mRNA shown in **d** and the homeostatic model assessment insulin resistance (HOMA-IR) index. Symbols represent individual patients control samples are shown in green, NAFLD are orange and NASH are red; Spearman's correlation test was used to test associations between TMEM127 and Insulin levels (shown in **e**) or HOMA-IR (shown in **f**). *$P < 0.05$; **$P < 0.01$; ***$P < 0.001$; **g** working model of Tmem127 function. The upper panel displays a summary of the effects of global loss of Tmem127 on insulin sensitivity and fat deposition under regular diet in young vs. adult mice or after high-fat diet (HFD). Adult knockout (KO) mice show higher hepatic and peripheral insulin sensitivity as well as diminished hepatic fat deposition even after a HFD challenge. The lower panel summarizes our findings in liver- or adipose-specific Tmem127 deficiency. Our data suggest that liver Tmem127 promotes hepatic glucose production and peripheral glucose uptake both by cell autonomous and non-cell autonomous pathways; fat Tmem127 inhibits adipogenesis potential and modulates hepatic gluconeogenesis. Furthermore, it is likely that Tmem127 in other tissues contribute to the metabolic phenotype. See also Supplementary Fig. 9A, B and Supplementary Tables 3 and 4. Source data are provided as a Source Data file

---

size measurement was performed according to the procedure described by Mul et al.[48] The average diameter of adipocyte cell from 4–5 fields of imaged H&E sections was analyzed using Image J.

**Fgf21 assays**. Plasma Fgf21 was measured from fed and 24 hr fasted mouse plasma using an ELISA kit (#RD291108200R; BioVendor) according to the manufacturer's instructions.

**Insulin assays**. Plasma insulin was measured from 6 h fasted mice or after stimulation with glucose using Milliplex MAP Mouse adipokine magnetic bead panel kit (#MADKMAG-71K, Millipore) according to the manufacturer's instructions.

**Triglyceride and free fatty-acid assays**. Triglyceride Assay Kit (#ab65336, Abcam) was used to measure tissue (liver) and plasma triglycerides and Free Fatty Acid Quantitation Kit (#MAK044, Sigma) was used to measure plasma free fatty acids, according to each manufacturer's instructions.

**Hepatic glucose production assay**. HepG2 TMEM127 KO or control cells were transferred to a 12-well plate at a density of $1 \times 10^6$ cells/well and cultured for 28 h incubated in glucose-free Dulbecco's modified Eagle's medium (Thermo Fisher Gibco, catalog no. D1050) supplemented with 2 mM-pyruvate(Sigma, catalog no. P2256) or phosphate-buffered saline (PBS) overnight, then Insulin 100 μM or PBS were added for 30 min and cells were harvested for RNA.

**Catecholamine assays**. Norepinephrine and epinephrine from plasma and adrenal tissue were measured using gas chromatography–mass spectrometry methods[49].

**Antibodies**. The following antibodies were obtained from Cell Signaling Technologies: phospho-S6 ribosomal protein/S235/236 (#2211), total S6 ribosomal protein (5G10) (#2217), phospho-AKT/S473 (#9271), phospho-AKT/T308 (#13038), total AKT (#9272), phospho-4EBP1 (##2855), Rictor (#2114), total ACACA/ACC (#3676), FASN (#3180), mTOR antibody for western (#2983) and immunoprecipitation (#2972); alpha-tubulin was from Sigma (#T9026); TMEM127 polyclonal antibody was from Bethyl laboratories (#A303-450A).

**Constructs**. The following plasmids were obtained from Addgene: SV40 Large-T antigen (Addgene #14088,), pCMV-VSV-G (#8454), pSPAX2 [12260], pMD2G [12259] pLentiCRISPRv2 [#52961], Rictor_1 shRNA (#1853) and pTM-LKO.1 (#8453).

**Reporting summary**. Further information on research design is available in the Nature Research Reporting Summary linked to this article.

## Data availability

Correspondence and requests for materials related to this study should be sent to dahia@uthscsa.edu. All data supporting the findings of this study are available within the paper and its Supplementary Information and as source data files, or from the corresponding author on reasonable request.

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

## Acknowledgements

We thank Nick Musi and Adam Salmon for insightful discussions, Martin Javors, Elizabeth Fernandez, Veronica Galvan, Yuji Ikeno, and Kat Fischer for access to technical resources; Myrna Garcia, Eric Baeuerle, An-Ping Lin, Jiyoon Ryu, Stacy Ann Hussong, Hak Joo Lee, Meenalakshmi M. Mariappan, Vanessa Martinez, and Vivian Diaz for technical support. This work was supported by the NIH/NIA T32AG021890 (S.S.); NIH/NIGMS GM114102 (P.L.M.D.), NIH/NIDDK DK102965 (L.Q.D.), Cancer Prevention and Research Institute of Texas (CPRIT) Individual Investigator Grants RP101202 and RP140473 (P.L.M.D), RP170146, RP190043 (R.C.T.A) and Training Grant RP140105 (Y.D.); Department of Defense CDMRP W81XWH-12-1-0508 (P.L.M.D.), Leukemia and Lymphoma Society TRP-6524-17, and VA MERIT—I01 BX001882-08 (R.C.T.A.). Animal studies were also supported by the San Antonio Nathan Shock Center, and the CTSA-IIMS (NIH/NCATS Grant UL1 TR001120 and UL1 TR002645). The content is solely the responsibility of the authors and does not necessarily represent the official views of the NIH.

## Author contributions

Conceptualization, methodology, and validation: S.S., P.L.M.D.; investigation: S.S., Y.D., Z.-M.C., A.L., Y.Q., Q.G., G.M.D., S.T., X.Z., N.H., C.S., M.F., Z.L., R.L.R.; resources and technical advice: B.S.K., S.H., S.A., L.Q.D., M.A.-G., L.N., R.C.T.A; writing, review and editing: S.S., R.C.T.A, P.L.M.D.; funding acquisition: R.C.T.A, P.L.M.D.; supervision: P.L.M.D.

## Competing interests

The authors declare no competing interests.
