## [Peer Review File · Nature Communications]

Reviewers' comments:

Reviewer #1 (Remarks to the Author):

This paper by Srikantan et al describes the metabolic phenotype of TMEM127 deficient mice. TMEM127 is known in humans, to be a tumor suppressor gene predisposing to pheochromocytomas and rare cases of renal cancers. As for other tumor suppressors, Tmem KO in mouse does not reproduce the clinical presentation of patients with these mutations. Instead, the authors report a comprehensive characterization of the metabolic phenotype of the mice, and decipher this complex issue using tissue-specific Tmem127 KO mice, in liver and adipocytes. Although these studies seem to have been well performed, they raise some issues on the relevance of data generated from genetically engineered mouse models in human physiopathology.

To address this point, I have several questions and/or suggestions:

1 Regarding the absence of an adrenal phenotype, the authors state that adrenal of the animal are normal as well as catecholamines secretion. Has mTOR signaling been studied in these tissues? Is the absence of adrenal phenotype due to the absence of mTOR activation, or is activation of this pathway insufficient to promote tumor growth in murine models?

2 Patients with pheochromocytomas often present a metabolic syndrome or diabetes that are caused by catecholamine secretion. The authors have studied patients with these mutations for a long time. Do they have any notion that this phenotype is somehow different in TMEM127 mutated patients. Patients being heterozygous for the TMEM127 mutation in normal cells, this would mean that haploinsufficiency would be sufficient to mediate a physiological response. In that view, what is the phenotype of Tmem127^{+/-} mice?

3 Numerous genetic studies in very large cohorts of patients have been published in type II diabetic patients. Were TMEM127 variants identified in these studies?

Minor point: authors should check the nomenclature of gene/protein in mouse and modify the text accordingly.

Reviewer #2 (Remarks to the Author):

To investigate the role of the tumor suppressor Tmem127, the authors generated the systemic Tmem127 deficient (Tmem127KO) mice. The body, adipose, and liver weights were significantly reduced in the Tmem127KO mice. The Tmem127KO mice showed an increased insulin sensitivity and glucose tolerance. Although the lipogenesis-associated genes were elevated in adipose tissue, the gluconeogenic- and lipogenesis-associated genes were decreased in livers of Tmem127KO mice. Next, the authors generated the liver-specific and adipose-specific Tmem127KO mice, respectively. No differences in the body and liver weight were observed in the liver-specific and adipose-specific Tmem127KO mice. The liver-specific Tmem127KO mice exhibited an increased insulin sensitivity, whereas the adipose-specific Tmem127KO mice showed the insulin resistance. However, the hepatic TG content and lipogenesis-associated genes were increased in the liver-specific Tmem127KO mice. The phosphorylation of Akt due to increased Rictor were promoted in the liver-specific Tmem127KO mice. Moreover, the tmem127 gene was significantly increased in high-fat diet-fed mice. Although the rate of body weight gain was significantly greater, the insulin sensitivity was reduced in the high-fat fed Tmem127KO mice.

Although this manuscript is of much interest, it is insufficient to lead to a certain conclusion.

1. To determine the insulin sensitivity in the organs (liver, adipose, skeletal muscle) of the Tmem127KO mice, the authors should conduct the clamp study or Akt phosphorylation experiment after insulin infusion.

2. Why did the Tmem127KO mice show an increased energy expenditure? Is energy expenditure also elevated in the liver-specific and adipose-specific Tmem127KO mice?

3. Since the body weight remained unchanged in the liver-specific and adipose-specific Tmem127KO mice, the reduced body weight seen in the Tmem127KO mice seemed attributable to other organs. Please discuss it in detail.
4. The authors should clarify whether the gluconeogenesis is increased or inhibited in the liver-specific Tmem127KO mice by using the clamp.
5. When the high-fat diet was fed in the liver-specific and adipose-specific Tmem127KO mice, did the mice exhibit improved insulin sensitivity and glucose tolerance? Does Rictor and the Akt phosphorylation increase in the high-fat diet-fed liver-specific Tmem127KO mice?
6. What mechanisms do the authors believe increased the Tmem127 expression levels in the high-fat diet fed mice and adult mice? Please clarify by some experiments.
7. Were the Tmem127 expression levels reduced in adipose tissue of high-fat fed mice?
8. Although the authors describe the decrease of the hepatic TG content in the adipose-specific Tmem127KO mice (Page 11 line9), hepatic TG content does not appear to be different between the control and adipose-specific Tmem127KO mice (Fig. 4I).
9. As to Fig. 4J and 4K, please add the Flx data (Control data).
10. Does the Tmem127 directly inhibit Rictor expression levels?
11. In page 6 line 12, do the authors mean Fig. S1G, instead of Fig. S1F? And in page 12 line 9, do they mean Fig. M, instead of Fig. 4C?
12. By what mechanisms do the authors believe insulin resistance is caused in the adipose-specific Tmem127KO mice?

Point-by-point response to the reviewers' comments.

Reviewer 1:

Although these studies seem to have been well performed, they raise some issues on the relevance of data generated from genetically engineered mouse models in human physiopathology.

To address this point, I have several questions and/or suggestions:

We thank the reviewer for his/her careful review, and for the insightful comments and questions.

1 Regarding the absence of an adrenal phenotype, the authors state that adrenal of the animal are normal as well as catecholamines secretion. Has mTOR signaling been studied in these tissues? Is the absence of adrenal phenotype due to the absence of mTOR activation, or is activation of this pathway insufficient to promote tumor growth in murine models?

This is a very relevant point. We have analyzed mTOR signaling in adrenals of *Tmem127* KO and found no substantial difference in the levels of phosphorylated mTOR target S6kinase (S6K) or 4EBP1 between age and gender-matched KO and WT mice. We have added a new figure (S1G) displaying representative results. As these animals had no evidence of pheochromocytoma (i.e. normal circulating or adrenal catecholamine levels and lack of detectable adrenal tumors), at this point we cannot distinguish whether the lack of tumor development in the *Tmem127* KO mice is due to insufficient level of mTOR activation in the adrenals, or whether other pathways might be involved in these tissues.

It is worth noting that similar to the *Tmem127* KO, other mouse models of pheochromocytoma susceptibility genes do not spontaneously develop pheochromocytomas, e.g. *Vhl*, *Sdhb*, *Sdhd*, *Fh* KOs, even when these models were designed to overcome early lethality by conditional (tissue-specific) expression (reviewed in Lussey-Lepoutre et al, Cell Tissue Res, 2018, <https://doi.org/10.1007/s00441-018-2797-y>), possibly suggesting that the mouse adrenal is 'protected' against some vulnerabilities observed in human disease. Therefore, it is possible that additional requirements (or additional genetic events) may be needed to recapitulate the human pheochromocytoma phenotype in the mouse.

2 Patients with pheochromocytomas often present a metabolic syndrome or diabetes that are caused by catecholamine secretion. The authors have studied patients with these mutations for a long time. Do they have any notion that this phenotype is somehow different in *TMEM127* mutated patients.

The reviewer raises an important issue. The glucose profile of patients carrying *TMEM127* mutations (and therefore with a germline heterozygous status) has not been systematically recorded. We have not documented any case of diabetes in the patients that we have directly followed up. Of note, diagnosis of diabetes in patients carrying *TMEM127* mutations would, in theory, be inconsistent with our findings in the mouse model of *Tmem127* deficiency, a genotype 'protective' of diabetes. However, as an impaired glucose tolerance profile occurs in 15–35% of patients with pheochromocytoma, we caution that the number of patients under observation is small, thus preventing us from drawing definitive conclusions. Clearly, detailed prospective studies designed to evaluate glucose and insulin tolerance of mutation carriers in a more systematic manner is warranted.

Patients being heterozygous for the *TMEM127* mutation in normal cells, this would mean that haploinsufficiency would be sufficient to mediate a physiological response. In that view, what is the phenotype of *Tmem127*^{+/-} mice?

Heterozygous mice do not differ from WT *Tmem127* with respect to body weight, body composition from birth until adulthood, the age range for which our data were analyzed. In addition, we did not detect pheochromocytomas by autopsy of heterozygotes (nor in homozygous KO mice). As we have not performed extensive analysis on these mice, it is possible that a subtler phenotype may be revealed by more detailed studies. Nonetheless, these observations agree with lack of overt phenotype, eventual diagnosis of pheochromocytoma notwithstanding, of *TMEM127* mutation carriers.

3 Numerous genetic studies in very large cohorts of patients have been published in type II diabetic patients. Were *TMEM127* variants identified in these studies?

This is an interesting point and we thank the reviewer for this insightful suggestion. Given the rarity of *TMEM127* pathogenic variants in humans (present in about 2% of pheochromocytoma patients- Yao et al, *JAMA*. 2010; 304(23):2611-2619. doi:10.1001/jama.2010.1830; Bausch et al, *JAMA Oncol*. 2017;3(9):1204-1212. doi:10.1001/jamaoncol.2017.0223), we did not anticipate that we would be able to detect *TMEM127* variants that might be significantly enriched in patients with diabetes, and indeed that was the case (for example, from available large cohorts of T2DM: <https://www.nature.com/articles/ng.2882>; <https://www.nature.com/articles/nature12828>). However, following the reviewer's suggestion and considering our findings that loss of liver *Tmem127* in mice leads to insulin protection and reduced hepatic fat deposition, we sought to evaluate whether *TMEM127* **expression** (instead of sequence variation) might be associated with human disease. To that end, we collaborated with Drs. Stephen Harrison and Sunil Ahuja, who have collected and analyzed liver biopsies from a cohort of patients with various degrees of non-alcoholic fatty liver disease (NAFLD), including advanced forms (NASH, nonalcoholic steatohepatitis), both of which are conditions associated with type 2 diabetes and insulin resistance. This is a well-annotated human cohort for which several metabolic parameters (glucose, insulin, BMI, HOMA-IR), as well as RNAseq from liver biopsies have been obtained. We have now incorporated these findings into our manuscript (subsection entitled ***Tmem127* expression correlates with insulin resistance states and fatty liver disease in mice and humans**, pages 13-15 and Figures 9E,9F,9G and Suppl Table 3 and 4). In summary, these additional data indicate that hepatic *TMEM127* expression in human liver correlates with fatty liver disease and is highest in patients with advanced forms of the disease (NASH, $p < 0.021$). Moreover, in these patients *TMEM127* expression is significantly associated with two parameters associated with insulin resistance: serum insulin levels ($r = 0.39$, $p = 0.005$) and HOMA-IR ($r = 0.35$, $p = 0.011$). These data provide substantial support to our observations in the murine models and strengthen the translational relevance of our findings.

Minor point: authors should check the nomenclature of gene/protein in mouse and modify the text accordingly.

We thank the reviewer for pointing this out. We have reviewed the manuscript and made corrections to reflect accurate mouse nomenclature versus human nomenclature for gene and protein throughout the text and figure legends.

Reviewer #2:

Although this manuscript is of much interest, it is insufficient to lead to a certain conclusion.

We appreciate the reviewer's thorough analysis of our work, and his/her acknowledgment of the relevance of our findings. Our experiments using the three mouse models, diet stress as well as multiple mouse and human pathogenic insulin resistant states have led us to conclude that *Tmem127* plays a role in glucose and lipid homeostasis. Specifically, our data suggest that liver *Tmem127* promotes hepatic gluconeogenesis and inhibits peripheral glucose uptake, while adipose *Tmem127* inhibits adipogenesis and signals to the liver to control hepatic gluconeogenesis. Further, as we found that mTORC2 is activated in *TMEM127*-deficient liver cells and furthered by suggestion that Rictor may modulate *Tmem127* levels, we propose that mTORC2 and *TMEM127* may influence each other's expression/activity to control insulin physiology through hepatic and adipose signals. Surely, the intricacies of the cross talk between mTORC2 and *TMEM127* remain to be defined, but with the mouse models that we generated and report for the first time herein, this becomes a feasible goal. Importantly, this work also provides insights into *Tmem127* regulation, which have translational relevance: hepatic *Tmem127* expression is modulated by metabolic status, being higher in high-fat diet-induced or genetically determined insulin resistant states, and, conversely, reversing to normal levels by diet or treatment with the insulin sensitizer pioglitazone. Importantly, in humans, liver *TMEM127* expression correlates with hepatic steatosis (NAFLD), steatohepatitis (NASH) and insulin resistance. Our results suggest that *TMEM127* may be a relevant target in insulin resistance. Although future studies will be needed to map all the molecular underpinnings of *TMEM127* action and regulation, its role in other metabolically active tissues, and under additional stress conditions, our studies provide the first insight into a hitherto unappreciated role of *TMEM127* in metabolic homeostasis.

These main conclusions of our work have been summarized in our abstract and in the discussion.

1. To determine the insulin sensitivity in the organs (liver, adipose, skeletal muscle) of the *Tmem127*KO mice, the authors should conduct the clamp study or Akt phosphorylation experiment after insulin infusion.

This is was an excellent suggestion. We have examined Akt phosphorylation after insulin injection and found substantially increased levels of Akt in liver and muscle of the KO compared to WT mice. These data, shown in new Fig. 4G, strengthen our findings that global loss of *Tmem127* in mice lead to improved insulin sensitivity.

2. Why did the *Tmem127*KO mice show an increased energy expenditure? Is energy expenditure also elevated in the liver-specific and adipose-specific *Tmem127*KO mice?

Based on our investigations of the higher resting metabolic rate (RMR) of *Tmem127* KO we were able to exclude increased catecholamines, increased thermogenesis, increased locomotor activity, activated lipolysis or beta-oxidation expression programs as potential causes for their energy expenditure. Thus, the mechanism for the higher energy expenditure of these mice, which is likely multifactorial, will require additional studies to be fully clarified. In attention to the reviewer comment, we evaluated the energy expenditure of the AKO mice and found that they have low respiratory quotient (RQ, new Fig 7J), but no change in RMR (new Suppl Fig S7C). These data suggest that the AKO, different from the whole-body KO,

preferentially uses fat as a source of energy under a chow diet. Unfortunately, we did not have sufficient numbers of age-matched liver specific *Tmem127* KO to further expand on these new data.

3. Since the body weight remained unchanged in the liver-specific and adipose-specific *Tmem127*KO mice, the reduced body weight seen in the *Tmem127*KO mice seemed attributable to other organs. Please discuss it in detail.

This is another very good point. We agree with the reviewer that the phenotype of the whole-body *Tmem127* KO mice likely reflects the combined effect of other tissues besides liver and adipose. We cannot rule out that *Tmem127* in hypothalamus, pancreatic beta cell, muscle or macrophages contribute to the body weight phenotype and their role should be assessed in future studies. As the *Tmem127*KO mice had low insulin levels, it is possible that this may have also contributed to the reduced body weight. These considerations have been incorporated in the discussion of the revised manuscript (Discussion).

4. The authors should clarify whether the gluconeogenesis is increased or inhibited in the liver-specific *Tmem127*KO mice by using the clamp.

The reviewer raises an important point. Our experiments indicate that *Tmem127* loss leads to attenuated hepatic gluconeogenesis. This conclusion is based on multiple lines of evidence from in vivo and in vitro observations, from the original version as well as new studies added to the revised manuscript: 1) KO mice had improved glucose clearance in ITT performed after overnight fasting, a condition which induces hepatic gluconeogenesis (Fig4C); 2) KO mice had lower increase in glucose levels in response to pyruvate (pyruvate tolerance test, Fig4B); 3) KO markedly reduced post-fasting liver expression of gluconeogenic genes, *G6pase* and *Pepck1* (Fig4D, new data); 4) in vitro pyruvate hepatic glucose synthesis in HepG2 cell lines lacking *Tmem127* (by CRISPR-Cas9) show reduced expression of gluconeogenic gene (Fig8B, new data). Taken together, these data suggest the presence of diminished hepatic gluconeogenesis, and are in agreement with the insulin-favorable profile of the whole body *Tmem127* KO and support hepatic insulin sensitivity. Importantly, the in vitro experiments indicate that these effects are liver autonomous. We acknowledge that clamp studies will be relevant to provide further insights into the dynamic of glucose metabolism in vivo in the liver-specific and adipose-specific KO, and we plan to perform these analyses in future studies.

5. When the high-fat diet was fed in the liver-specific and adipose-specific *Tmem127*KO mice, did the mice exhibit improved insulin sensitivity and glucose tolerance? Does Rictor and the Akt phosphorylation increase in the high-fat diet-fed liver-specific *Tmem127*KO mice?

The LKO and AKO have not yet been exposed to a high-fat diet, but we recognize that these are important questions to address in the future. In attention to the reviewer comment, and since we observed changes in insulin sensitivity at regular diet conditions, we have evaluated the AKT phosphorylation in liver-specific KO (*Tm-LKO*) and adipose-specific KO (*Tm-AKO*) chow-fed mice: we found it to be modestly increased in the *Tm-LKO* (new Fig7B), consistent with the insulin sensitivity observed in this model (Fig6G), but not significantly altered in the *Tm-AKO* (new Suppl FigS7A). Rictor expression levels were variable but not significantly different from the control mice under these conditions (new Suppl FigS7B).

6. What mechanisms do the authors believe increased the *Tmem127* expression levels in the high-fat diet fed mice and adult mice? Please clarify by some experiments.

The mechanisms regulating Tmem127 expression are not well established, in particular because this gene promoter has not yet been characterized. Nonetheless, our work has uncovered a number of conditions that upregulate hepatic Tmem127 transcription, including older age (as opposed to younger, Fig3G), high-fat diet (Fig9A), genetic models of leptin deficiency, Ob/Ob (Fig9B), Db/Db (Fig9C). These are all conditions in which there is concomitant association of increased body mass and increased insulin levels/insulin resistance. In contrast, we found that Tmem127 expression is downregulated after body weight reduction (switch from a HFD to chow-Fig9A) or improved glucose/insulin tolerance (after pioglitazone treatment, Fig 9C). Our in vitro data suggest that insulin exposure increases TMEM127 levels (Figs 8A, 8E), suggesting that insulin modulates TMEM127. Most of these changes occur at the mRNA level thus implying a transcriptional mechanism as a central regulator of Tmem127 abundance. However, additional investigations will be needed to fully establish the molecular mechanisms of TMEM127 expression and whether these changes result from a direct effect, or are indirect, as a result of activation/inactivation of other factor(s)/condition(s).

7. Were the Tmem127 expression levels reduced in adipose tissue of high-fat fed mice?

This is an important point to have clarified, and in attention to the reviewer comment we examined it. We found that changes in expression of Tmem127 in adipose tissue varied depending on the type of fat depot: Tmem127 expression was lower in iWAT (Suppl Fig 8A) but higher in eWAT (Suppl Fig 8B) after high-fat diet compared to chow-fed mice, suggesting a complex regulation in fat tissue will need to be further examined.

8. Although the authors describe the decrease of the hepatic TG content in the adipose-specific Tmem127KO mice (Page 11 line9), hepatic TG content does not appear to be different between the control and adipose-specific Tmem127KO mice (Fig. 4I).

The reviewer is correct, and we apologize for not having clearly indicated the lack of statistical difference in the graph. Although there is a trend toward decreased hepatic Tg levels in the Tm-AKO mice, the high degree of variability (SD) between individual samples in this group possibly contributed to the lack of statistical significance. We have now added a label showing the lack of statistical difference between Flx and Tm-AKO (Suppl FigS6C).

9. As to Fig. 4J and 4K, please add the Flx data (Control data).

These graphs previously displayed the data as a ratio between liver-specific KO and Flx (control) or adipose-specific KO and Flx (control). For clarity, we have now modified the display to show the Flx data side by side with the KO data. Figures 7C and 7D for the LKO, and figures 7F and 7G for the AKO.

10. Does the Tmem127 directly inhibit Rictor expression levels?

We have not found a direct effect of Tmem127 on Rictor expression, as shown in the blots of whole-body Tmem127 KO (Fig.4F, new Fig.S7B), HepG2 cells (Fig.8E) or LKO and AKO liver (new Fig.S7B). However, interestingly, we noticed that Rictor status influenced endogenous Tmem127 protein levels (Fig. 8E), suggesting that Tmem127 may actually be regulated by Rictor and/or mTORC2. Additional studies will be required to characterize the nature of the interaction between Tmem127 and Rictor.

11. In page 6 line 12, do the authors mean Fig. S1G, instead of Fig. S1F? And in page 12 line 9, do they mean Fig. M, instead of Fig. 4C?

We apologize for the confusion. Yes, the reviewer is correct. We note that in response to the reviewer's comments and the new experiments included, the manuscript has been extensively revised, and some figures have been renumbered. We have corrected and revised the figure numbers accordingly- figure S1G is now S1H; and figure 4M is now Fig.8C.

12. By what mechanisms do the authors believe insulin resistance is caused in the adipose-specific Tmem127KO mice?

Our data suggest that adipose Tmem127 signals to the liver to inhibit hepatic gluconeogenesis. This stems from the observation that mice lacking adipose Tmem127 have an inappropriate regulation of hepatic gluconeogenesis. In these mice, the levels of gluconeogenic gene expression during the fed state (when hepatic gluconeogenesis is inhibited) are comparable to those detected in conditions of fasting, when hepatic gluconeogenesis is activated. These observations suggest that the insulin resistance observed in the adipose-specific Tmem127 KO is due to increased hepatic glucose production (these observations are now included in the discussion and summarized in Fig9H). Tissue-specific antagonistic effects in metabolism have been reported in other models and reflect complex feedback mechanisms and tissue cross-talk, both at physiological and stress conditions (e.g. Lxr model Beaven et al <https://www.sciencedirect.com/science/article/pii/S1550413113002465>). Therefore, additional in vivo studies under dietary stress will be required to provide additional insights into these processes and to precisely quantify both liver glucose production and peripheral tissue glucose uptake. The specific molecular mechanisms underlying the actions of Tmem127 are currently not fully defined. However, as we found that mTORC2 is activated in the Tmem127 deficient liver cells and, in turn, that Rictor may modulate Tmem127 levels, we posit that mTORC2 is an important mediator of the metabolic effects of Tmem127. Additional investigations to precisely delineate the molecular underpinnings of the interplay between TMEM127 and mTORC2 in regulating metabolic homeostasis will be required. For example, an area of focus for future examinations will be lysosomal positioning, a phenomenon recently reported to be required for mTORC2 activity (<https://www.sciencedirect.com/science/article/pii/S1097276519303624>), and which we previously described to be disrupted in cells deficient in Tmem127 (Deng et al, <https://www.sciencedirect.com/science/article/pii/S1097276519303624>).

REVIEWERS' COMMENTS:

Reviewer #1 (Remarks to the Author):

The authors have addressed the questions and comments I had raised.

Reviewer #2 (Remarks to the Author):

The authors have done a satisfactory job in addressing the previous concerns of this reviewer.